# Multiscale analysis reveals that diet-dependent midgut plasticity emerges from alterations in both stem cell niche coupling and enterocyte size

**Alessandro Bonfini[1], Adam J Dobson[2], David Duneau[3,4], Jonathan Revah[1], Xi Liu[1], Philip Houtz[1], Nicolas Buchon[1]***

[1]Cornell Institute of Host-Microbe Interactions and Disease, Department of Entomology, Cornell University, Ithaca, United States; [2]Institute of Molecular, Cell and Systems Biology, University of Glasgow, Glasgow, United Kingdom; [3]Université Toulouse 3 Paul Sabatier, CNRS, UMR5174 EDB (Laboratoire Évolution & Diversité Biologique), Toulouse, France; [4]Instituto Gulbenkian de Ciência, Oeiras, Portugal

**\*For correspondence:**
nicolas.buchon@cornell.edu

**Competing interest:** The authors declare that no competing interests exist.

**Abstract** The gut is the primary interface between an animal and food, but how it adapts to qualitative dietary variation is poorly defined. We find that the *Drosophila* midgut plastically resizes following changes in dietary composition. A panel of nutrients collectively promote gut growth, which sugar opposes. Diet influences absolute and relative levels of enterocyte loss and stem cell proliferation, which together determine cell numbers. Diet also influences enterocyte size. A high sugar diet inhibits translation and uncouples intestinal stem cell proliferation from expression of niche-derived signals, but, surprisingly, rescuing these effects genetically was not sufficient to modify diet's impact on midgut size. However, when stem cell proliferation was deficient, diet's impact on enterocyte size was enhanced, and reducing enterocyte-autonomous TOR signaling was sufficient to attenuate diet-dependent midgut resizing. These data clarify the complex relationships between nutrition, epithelial dynamics, and cell size, and reveal a new mode of plastic, diet-dependent organ resizing.

## Introduction

Nutrition is a principal determinant of animal health and fitness, affecting aging, metabolic disease, and fecundity (*López-Otín et al., 2016*). Understanding how diet impacts physiology has broad societal implications, especially at a time when the average body mass index of human populations is continuously increasing and weight is a critical risk factor for disease (*Finkelstein, 2014*; *Kebede and Attie, 2014*). The impact of diet on health is evolutionarily conserved: in *Drosophila melanogaster* and other organisms such as mice and zebrafish, dietary restriction has been extensively studied, revealing an inverse relationship between lifespan and caloric content, as well as fecundity (*Piper and Bartke, 2008*; *Simpson and Raubenheimer, 2012*). Interestingly, the relationship of lifespan and reproduction has been shown to depend not only on quantity (i.e., calories), but also on the relative proportions of certain nutrients (e.g., amino acid imbalance; *Grandison et al., 2009*; *Solon-Biet et al., 2019*; *Solon-Biet et al., 2014*). This suggests that organismal physiology is influenced not only by quantity of food, but also by qualitative changes to dietary composition (*Piper et al., 2017*).

For animals, the gut is the sole interface with ingested food, and as such is an important regulator of organismal physiology. *Drosophila* is no exception: most digestion and absorption occur in the endoderm-derived midgut. The *Drosophila* midgut is a regionalized, tubular epithelial monolayer

sheathed in visceral muscles (*Demerec, 1950*) and is akin in function to the mammalian intestine (*Apidianakis and Rahme, 2011*; *Liu et al., 2017*). In *Drosophila*, five main midgut regions have been described, which can be grouped for simplicity as an anterior midgut dedicated to digestion (regions 1 and 2), an acidic middle midgut (region 3), and a posterior midgut specialized for absorption (regions 4 and 5) (*Buchon et al., 2013*). The midgut epithelium is mostly composed of a population of mature absorptive cells called enterocytes (ECs), which is maintained by proliferative intestinal stem cells (ISCs) differentiating through a transient enteroblast (EB) phase. ISCs can also differentiate into enteroendocrine cells (EEs) through a pre-EE progenitor phase (*Micchelli and Perrimon, 2006*; *Ohlstein and Spradling, 2006*; *Zeng and Hou, 2015*). The ISCs are distributed throughout the epithelium and have the ability to divide either asymmetrically (giving rise to one new ISC and one differentiated cell) or symmetrically (resulting in two identical progeny; *de Navascués et al., 2012*; *Micchelli and Perrimon, 2006*; *Ohlstein and Spradling, 2007*). Multiple pathways orchestrate ISC proliferation, either in response to changes in the nutritional environment or following infection-, abrasion-, or chemical-derived stresses (*Bonfini et al., 2016*; *Buchon et al., 2009b*; *Buchon et al., 2009a*; *Buchon et al., 2010*; *Jiang et al., 2009*). The TOR and insulin pathways regulate ISC proliferation and the growth of EBs and ECs on nutrient-rich diets (*Amcheslavsky et al., 2011*; *Choi et al., 2011*; *Haller et al., 2017*; *Kapuria et al., 2012*; *H-J et al., 2015*; *O'Brien et al., 2011*; *Strilbytska et al., 2017*; *Wen et al., 2017*). At the onset of the regenerative response, cytokines (Unpaired 2 and 3 [Upd2/3]) are released, resulting in the secretion of epithelial growth factors (e.g., Vein or Vn) by visceral muscles. Vein and other EGFs, together with EB-derived Wingless (Wg), initiate ISC proliferation via activation of the EGFR, JAK-STAT, and Wnt pathways (*Biteau and Jasper, 2011*; *Buchon et al., 2009a*; *Jiang et al., 2009*; *Zhou et al., 2013*). Under homeostatic conditions, the activity of these pro-mitotic pathways in ISCs is thought to be coupled with the expression levels of ligands secreted from the niche, ultimately determining ISC behavior (*Liang et al., 2017*). Both the frequency and the type (symmetrical vs. asymmetrical) of ISC mitosis are thought to underlie the dynamic response of the midgut tissue to diverse physiological conditions. To contextualize these signaling pathways, we must understand how their regulation varies in distinct physiological conditions – such as on different diets – and characterize how their variable outputs at the cellular level scale to the growth of specific regions and the whole gut.

Organ growth has been studied since the 1930s (*Penzo-Méndez and Stanger, 2015*; *Twitty, 1930*; *Twitty and Schwind, 1931*), mostly in the context of development. In *Drosophila*, the study of imaginal discs (*Bryant and Levinson, 1985*; *Gokhale and Shingleton, 2015*; *Neufeld et al., 1998*) revealed that both cell number and cell size contribute to the final size of the adult organ (*Neufeld et al., 1998*). In imaginal discs, cells divide and increase in number until the organ reaches a set size. Strikingly, in this system cell growth is dominant over, and compensates for, cell division defects (*Neufeld et al., 1998*). However, not all modes of growth regulation can be understood based on these developmental models. For instance, adult organs such as the intestine can still grow in adults but can also shrink in response to stimuli: the adult midgut can reshape itself, with bouts of organ shrinkage followed by regrowth in response to both damage (*Buchon et al., 2010*; *Buchon et al., 2009b*; *Buchon et al., 2009a*; *Jiang and Edgar, 2009*) and nutrient availability (*O'Brien et al., 2011*). Similar adaptive intestinal recovery after fasting and refeeding has been described in vertebrate models, highlighting the strong evolutionary conservation of the intestine's functional response to nutrient availability (*Tamaoki et al., 2016*; *ÖH et al., 2012*). As the midgut epithelium faces routine shedding of epithelial cells, regulation of adult midgut size fundamentally differs from the imaginal disc because its size can be determined by not only gain but also loss of cells. Therefore, to complete our understanding of how organ size is regulated, we must integrate new information on the balance of cell gain and loss, and the *Drosophila* midgut is an ideal model to investigate these questions.

Epithelial dynamics in the *Drosophila* midgut are described by two complementary models that capture the turnover of cells. A first model assumes that, in homeostatic conditions, the constant loss of mature epithelial cells is compensated by ISC proliferation, which is regulated by a feedback loop of pro-mitotic signals from dying ECs and the ISC niche (*Buchon et al., 2009b*; *Buchon et al., 2009a*; *Jiang et al., 2009*; *Liang et al., 2017*; *Micchelli and Perrimon, 2006*; *Ohlstein and Spradling, 2007*). In this model, cell gain is mechanistically coupled to cell loss, ensuring homeostasis. A second model allows for the midgut to respond to the presence of food. Such adaptive growth, presumably underpinned by changes in ISC activity and their mode of division (*O'Brien et al., 2011*), is thought

to balance the costs of maintaining a midgut against the benefits of acquiring nutrients from food. This model implies that the coupling between cell loss and ISC proliferation is not maintained, thus allowing for changes in cell number and organ size. To date, evidence for each of these two models has relied on the assumption that measurements of ISC proliferation in specific regions capture epithelial dynamics throughout the whole midgut, and consequent regulation of organ size. In addition to epithelial dynamics, changes in EB and EC ploidy in response to diet or infection have been reported, but not integrated into models of midgut growth (*Choi et al., 2011*; *Xiang et al., 2017*). Since ploidy often correlates with cell size, and most epithelial mass comprises ECs, these findings suggest that EC size could be an additional factor determining midgut size. To fully characterize epithelial cell dynamics, global organ-scale measurements of both cell gain and loss, and how they vary, are needed.

In this study, we characterize the response of the midgut to dietary variation, integrating the behavior of stem cells and ECs into a unified model, and scaling from molecular and cellular effects to resizing of the whole organ. We show that qualitative dietary variation can regulate midgut size, determined by opposing effects of sugar and a panel of other nutrients. We also outline novel mechanisms that regulate midgut size, showing that a high-sugar, low-yeast diet decouples stem cell proliferation from pro-mitotic niche signals by inducing translational blockage. Organ-level quantifications of cell gain, cell loss, and cell size indicate that midgut resizing is an emergent property of these three aspects of epithelial dynamics. However, we find that the main driver of midgut size in response to nutrient quality is in fact EC size, which is regulated by autonomous TOR signaling, and can even compensate for deficiencies in stem cell proliferation. Altogether, these findings provide a new, integrative perspective on the environmental, cellular, and molecular regulation of tissue homeostasis.

## Results

### Diet composition affects overall size and regional allometry of the midgut

It was previously described that the *Drosophila* midgut requires food in order to properly develop after eclosion (*Choi et al., 2011*; *O'Brien et al., 2011*). Diet composition can also affect the size of the adult midgut (*Ponton et al., 2015*). As *D. melanogaster* feeds naturally on rotten fruits (*Kohler, 1993*), we manipulated sucrose and yeast, which are widely used in laboratory *Drosophila* media and considered representative of natural nutrient sources. To ensure we measured only responses to the adult diet and not developmental differences, larvae were raised on a common pre-experiment diet before being moved to experimental diets (*Figure 1A*). We chose two isocaloric diets, which differed only in the relative abundance of ingredients (*Figure 1B*), based on their degree of difference in midgut size despite being isocaloric (*Ponton et al., 2015*). A detailed description of the recipe for these diets (and all other diets used in this study) can be found in *Supplementary file 1*. One diet was rich in sucrose and low in yeast (high sugar [HS] diet), while the other was rich in yeast and low in sugar (high yeast [HY] diet) (*Figure 1B*). The midguts of wild-type mated female Canton-S (Cs) flies feeding on the HY diet were on average 31% longer (*Figure 1C–E*, *Video 1*) and 44% wider than the midguts of flies feeding on the HS diet (*Figure 1—figure supplement 1A*). We found that width measurements were variable along the midgut and often affected by the volume of the internal bolus and microscope slide compression, so we decided to focus on midgut length as proxy of overall size. For reference, compared to two 'standard' diets in the field, midgut size of flies on the HY diet resembled in size Bloomington cornmeal and Bloomington molasses (*Figure 1—figure supplement 1B*). We also tested whether differences in midgut length depended on sex and mating status since midgut size and stem cell behavior are affected by these parameters (*Ahmed et al., 2020*; *Reiff et al., 2015*; *White et al., 2021*). Indeed, we found that guts of unmated females and males were significantly less different between HS and HY diets than the ones of mated females (*Figure 1—figure supplement 1C*), so we continued to work with mated females in subsequent experiments. Since diet composition affects feeding rate (*Carvalho et al., 2005*), we asked if an increased food intake, and thus calories, could explain the observed increase in midgut length on the HY diet. Flies on HS fed more than flies on HY, suggesting that growth on HY is not due to increased ingestion (*Figure 1—figure supplement 1D*). Diet also has the potential to affect midgut microbes, which regulate midgut homeostasis (*Broderick et al., 2014*; *Buchon et al., 2009a*). We therefore repeated our experiments in germ-free conditions, finding that microbes were not required for the response of midgut length to diet (*Figure 1—figure*

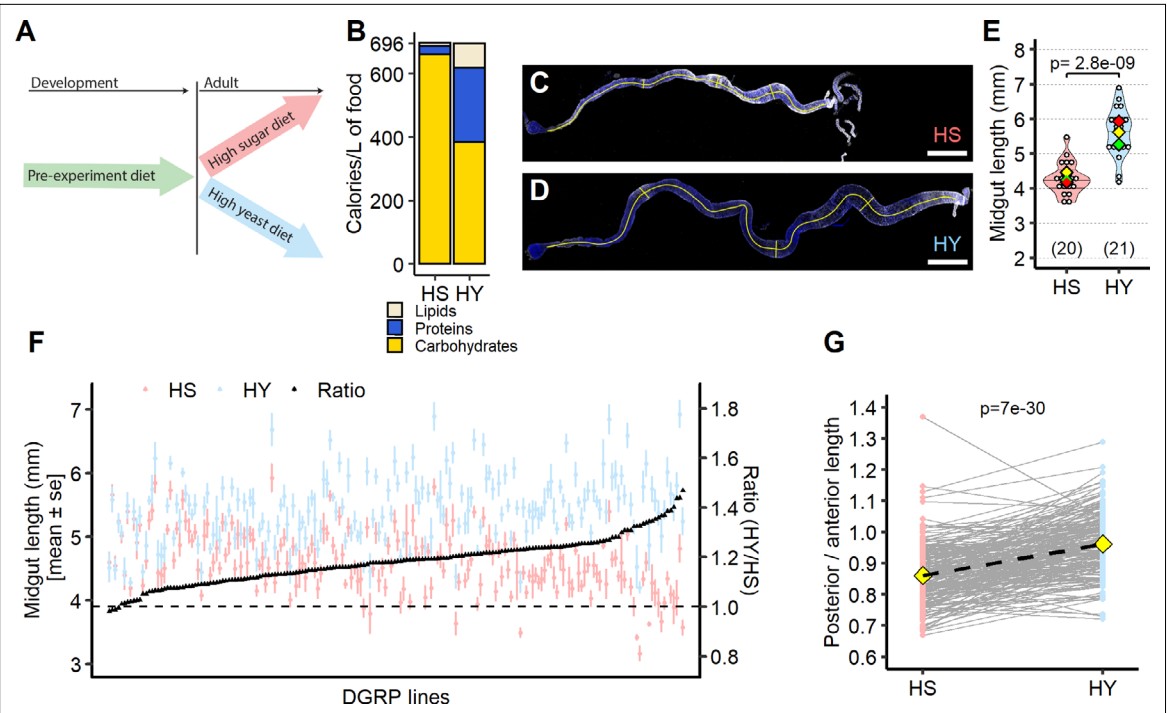

**Figure 1.** Diet composition affects size and regional allometry of the midgut. (**A**) Illustration of general dietary treatment design. Flies were reared on pre-experiment diet during development. At eclosion, flies were allocated to either high sugar (HS) or high yeast (HY) before midgut dissection at 5 days post eclosion. (**B**) Nutritional composition (proteins, carbohydrates, and lipids) of the two isocaloric diets used as a basis for this study as calories per liter of food: enriched in sugars (HS) or yeast (HY). (**C–E**) Canton-S (Cs) flies fed on HS diet (**C**) have shorter midguts than flies on HY (**D**). Quantification of midgut length for HS vs. HY at 5 days post eclosion (**E**). (**F**) Midgut length response to diet is strongly variable across the *Drosophila* Genetic Reference Panel (DGRP), with HY being generally longer than HS (i.e., the ratio length on HY/length on HS is between 1 and 1.4). (**G**) Midgut resizing is allometric between regions of the midgut. Posterior midguts of flies fed HY diet exhibit a greater increase than anterior regions. For the violin/dot plots shown in this figure, white dots represent single midgut measurements. Lozenges represent mean of repeats. Violin plots are color coded according to diets (HS = red, HY = light blue throughout the article). Numbers in parentheses at the bottom of charts indicate sample size. Additional information on the statistics can be found in *Supplementary file 2*. Scale bars are 500 µm for all images.

The online version of this article includes the following figure supplement(s) for figure 1:

**Figure supplement 1.** Diet composition affects midgut size independently of microbiota.

**Figure supplement 2.** Different regions of the midgut respond variably to diet composition.

**Figure supplement 3.** Genome-wide association identifies genes underlying natural variation in the midgut response to diet composition.

**Source data 1.** Numeric data for *Figure 1B*.

**Figure supplement 1—source data 1.** Numeric data for *Figure 1E* and *Figure 1—figure supplement 1A*.

**Source data 2.** Numeric data for *Figure 1F and G*.

**Figure supplement 1—source data 2.** Numeric data for *Figure 1—figure supplement 1B*.

**Figure supplement 1—source data 3.** Numeric data for *Figure 1—figure supplement 1C*.

**Figure supplement 1—source data 4.** Numeric data for *Figure 1—figure supplement 1D*.

**Figure supplement 1—source data 5.** Numeric data for *Figure 1—figure supplement 1E*.

**Figure supplement 3—source data 1.** Summary of GWAS analysis, shown in *Figure 1—figure supplement 3A*.

**Figure supplement 3—source data 2.** Complete result of GWAS analysis, shown in *Figure 1—figure supplement 3A*.

---

*supplement 1E*). These experiments demonstrate that dietary composition affects midgut size independently of gut microbes and the caloric content of diet.

In nature, most traits vary in magnitude due to genetic variation and because of genotype-by-environment variation (*Albert and Kruglyak, 2015*; *Timpson et al., 2018*). It is also known that individuals in a population vary in their physiological response to diet (*Garlapow et al., 2015*; *Jehrke et al., 2018*; *Uchizono and Tanimura, 2017*). We therefore wondered whether the effect of diet composition on midgut size that we saw with the Canton-S strain was variable and generalizable. We

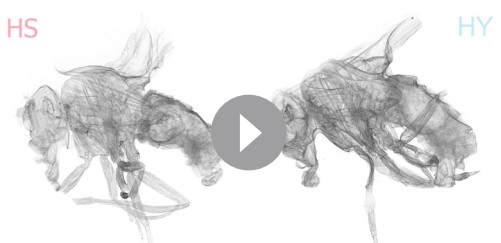

**Video 1.** Nano CT scan and rendering of internal organs. Video shows flies kept on high-sugar (HS) diet on the left, and flies kept on high-yeast (HY) diet on the right. At 5 s, a rendering of the ovaries appears in blue, showing increased size on HY diet. At 10 s, a rendering of the midgut (anterior and middle in light blue, posterior in green, and hindgut in purple) appears, showing increased size of the midgut on HY diet.

https://elifesciences.org/articles/64125/figures#video1

took the genetically diverse lines that constitute the *Drosophila* Genetic Reference Panel (DGRP, *Mackay et al., 2012*) and measured the impact of the HS and HY diets on midgut size, with three iterations of full experimental replication. In 184 out of 188 lines tested, flies feeding on HY had qualitatively larger midguts than flies feeding on HS (HY/HS ratio higher than 1, *Figure 1F*). The magnitude of response was genetically variable, with the fold-increase in size varying from 0.98 to 1.47 (median = 1.18). The response to diet was statistically significant in 132 lines, and we note that the 56 nonresponsive lines were found throughout the size distribution on either HS or HY. We found that ~15% of the variance in response to the diet was explained by genetic variation, which was lower than what was found for body size plasticity (between 33% and 52%; *Lafuente et al., 2018*). Collectively, these results confirmed that an elevated yeast to sucrose ratio generally promotes midgut size, but that this is a quantitatively variable trait (*Figure 1F*).

The digestive tract comprises multiple distinct and functionally specialized regions (*Buchon et al., 2013*; *Dutta et al., 2015*; *Marianes and Spradling, 2013*), as depicted in *Figure 1—figure supplement 2A*. We tested whether the diet-dependent change in midgut size acts uniformly on all regions of the *Drosophila* midgut by measuring the length of anterior, middle, and posterior midguts in the DGRP. Overall, all midgut regions were longer on the HY diet than the HS diet (*Figure 1—figure supplement 2B*). However, change in posterior midgut length predicted change in total length with 80% accuracy (i.e., $R^2 = 0.8$), while anterior and middle midgut explained 60 and 40% of total midgut length change, respectively (*Figure 1—figure supplement 2B*). In addition, the posterior midgut was more responsive to food than the anterior midgut or the entire midgut (*Figure 1—figure supplement 2B*). These results suggest that different midgut regions resize to a variable extent, and that diet composition affects the allometry of midgut regions. By quantifying the relative proportions of the anterior and posterior midguts of flies feeding on the HS and HY diets, we found that the length of the anterior midgut exceeds that of the posterior on the HS diet (*Figure 1G*). However, on the HY diet, the lengths of both regions are close to equal (*Figure 1G*). This suggests that the posterior midgut is more consistently responsive to changes in diet composition than the anterior midgut and that the relative proportions of the midgut regions change with diet. Altogether, our data demonstrate that diet composition affects the allometry of midgut regions, and that resizing is subject to genetic variation.

DGRP lines are fully genome-sequenced, with publicly available data on genetic polymorphisms (*Mackay et al., 2012*). We therefore sought to identify the genetic determinants underlying population variation in diet-dependent midgut resizing. We tested the association of the change in midgut length on HS and HY diets with ~1.9M genetic variants. In total, we identified 638 loci as strongly associated with response to diet ($p<10^{-9}$), including loci mapped in genes coding for proteins associated with cell junctions (intronic variants in *CadN*, *Nrg,* and *Magi*), cell division (5′ UTR variant in *insc* and modifier in *slik*), epigenetic regulation (missense variant in *Su(var)2-HP2*), and growth/differentiation (intronic variant in *tkv*) (*Figure 1—figure supplement 3A*). All these processes have been associated with tissue turnover, suggesting that changes in midgut length could result from altered cell dynamics in the midgut (*Chen et al., 2020*; *Hung et al., 2020*; *Izumi et al., 2012*; *Li et al., 2013a*; *Li et al., 2013b*; *Ma et al., 2019*; *Tian et al., 2017*). Thus, natural variation in the response of midgut size to dietary changes maps to genes involved in functions of probable relevance to organ growth.

## Sugar opposes yeast-induced increase in midgut size

Our results demonstrated that diet composition influences midgut size; however, it remained unclear whether the amount (or lack) of yeast and/or sucrose is responsible for this change or whether the

relative proportions of sucrose and yeast were responsible. To answer this question, we utilized a nutritional geometry approach (*Simpson and Raubenheimer, 1995*), which enables one to separate phenotypic impacts of relative versus total nutrient availability. We systematically varied the amounts of yeast and sucrose across 28 diets, studying five different ratios and four different caloric levels, with additional diets in points of interest in the diet space (*Figure 2—figure supplement 1A*). Nutritional geometry revealed yeast as a major driver of midgut length: increasing the amount of yeast in the diet increased midgut length. Interestingly, we also detected an opposite effect of sucrose: increasing levels of sucrose abrogated the growth-promoting effect of yeast (*Figure 2A*). To test the impact of total nutrient ingestion, we measured the amount of food ingested per diet in the same preparations of flies (*Figure 2—figure supplement 1B*), which allowed us to plot total midgut length over diet, normalized to food passage (*Figure 2—figure supplement 1C*). This revealed a large area of diet space devoid of points, precluding a meaningful surface-plot analysis, so we plotted midgut length over ingestion-normalized estimates of yeast, sugar, and yeast to sugar ratio (*Figure 2B–D*). This correlative approach showed that midgut length increased as a function of yeast ingested, before a plateau and then a slight decrease (*Figure 2B*), while increasing the amount of sucrose ingested decreased length (*Figure 2C*). Overall, midgut length seemed to be proportional to the yeast to sucrose ratio (*Figure 2D*), consistent with our conclusions that sucrose opposes yeast-induced growth.

Yeast, which appeared to drive midgut size, is a complex nutrient source, comprising ~45% proteins, ~40% carbohydrates, ~8% lipids, vitamins, and mineral traces. We asked whether a single nutrient class might account for the overall effect of yeast on midgut length by adding specific nutrients to a diet containing only the amount of sucrose found in HY, devoid of any yeast. These added nutrients were one of a source of proteins, amino acids, lipids, vitamins, and minerals. Surprisingly, no single nutrient recapitulated the effect of HY (*Figure 2E*). However, a combination of proteins, lipids, and vitamins promoted growth to a point that was statistically indistinguishable from midguts of flies fed HY (*Figure 2E*). We speculated that additional characteristics of the HY diet could be influencing midgut size, such as texture (*Li et al., 2018*). The addition of fibers to the diets (*Figure 2—figure supplement 2A*) did not affect midgut length, although altering density by varying agar content could affect midgut length (*Figure 2—figure supplement 2B*). However, none of these modifications were able to mimic the impact of yeast on midgut length. Together, these experiments suggest that a panel of yeast-derived nutrients is likely required to promote midgut growth, rather than a single nutrient, as is the case for other phenotypes, such as longevity and fecundity (*Grandison et al., 2009*).

To further test the opposite effect of sugar on midgut size, we analyzed the impact of diets with the caloric content of sucrose substituted with lipids. A lipids-only diet, isocaloric to HS and HY diets, resulted in short midguts (yeast:lipid 0:1, *Figure 2F*), in agreement with the hypothesis that yeast is required for growth. However, the addition of a small quantity of yeast (the same quantity found in the HS diet) to a lipid-based and sucrose-free diet led to midgut lengths comparable to those on HY diet (yeast:lipid 1:14, *Figure 2F*). These results demonstrate that in the absence of sucrose a small amount of yeast is sufficient for midgut growth, while on the HS diet (yeast:sugar 1:14), the same amount of yeast is not sufficient. This result confirms the opposing role of sucrose on yeast-induced growth.

Sucrose is one of many sugars, and itself a disaccharide of glucose and fructose. It was possible that one of these moieties alone may have been responsible for blocking the growth-promoting effect of yeast, or that this was an effect specific to disaccharides. We therefore compared the impacts of equivalent levels of sucrose, glucose, fructose, and maltose (a glucose disaccharide) on midgut size (*Figure 2G*). All four sugars decreased midgut length. To help distinguish between sensory and metabolic mechanisms regulating the size of the midgut, we also tested the effects of a nutritious, but not palatable sugar (sorbitol) and of a palatable, but not nutritious sugar (arabinose) on the size of the midgut (*Figure 2—figure supplement 2C*; *Burke and Waddell, 2011*). Response to sorbitol was similar to the response to sucrose, suggesting that sensory mechanisms are not key in defining the size of the midgut. On the other hand, HS diet made with arabinose was lethal (flies survived only 2–3 days), and HY diet made with arabinose reduced midgut size, suggesting some damaging effect of this sugar on either the midgut itself or organismal metabolism. Altogether, these data demonstrate that multiple nutrients from yeast collectively increase midgut size, including proteins, lipids, vitamins, and minerals. By contrast, sugars oppose the impact of yeast on midgut size.

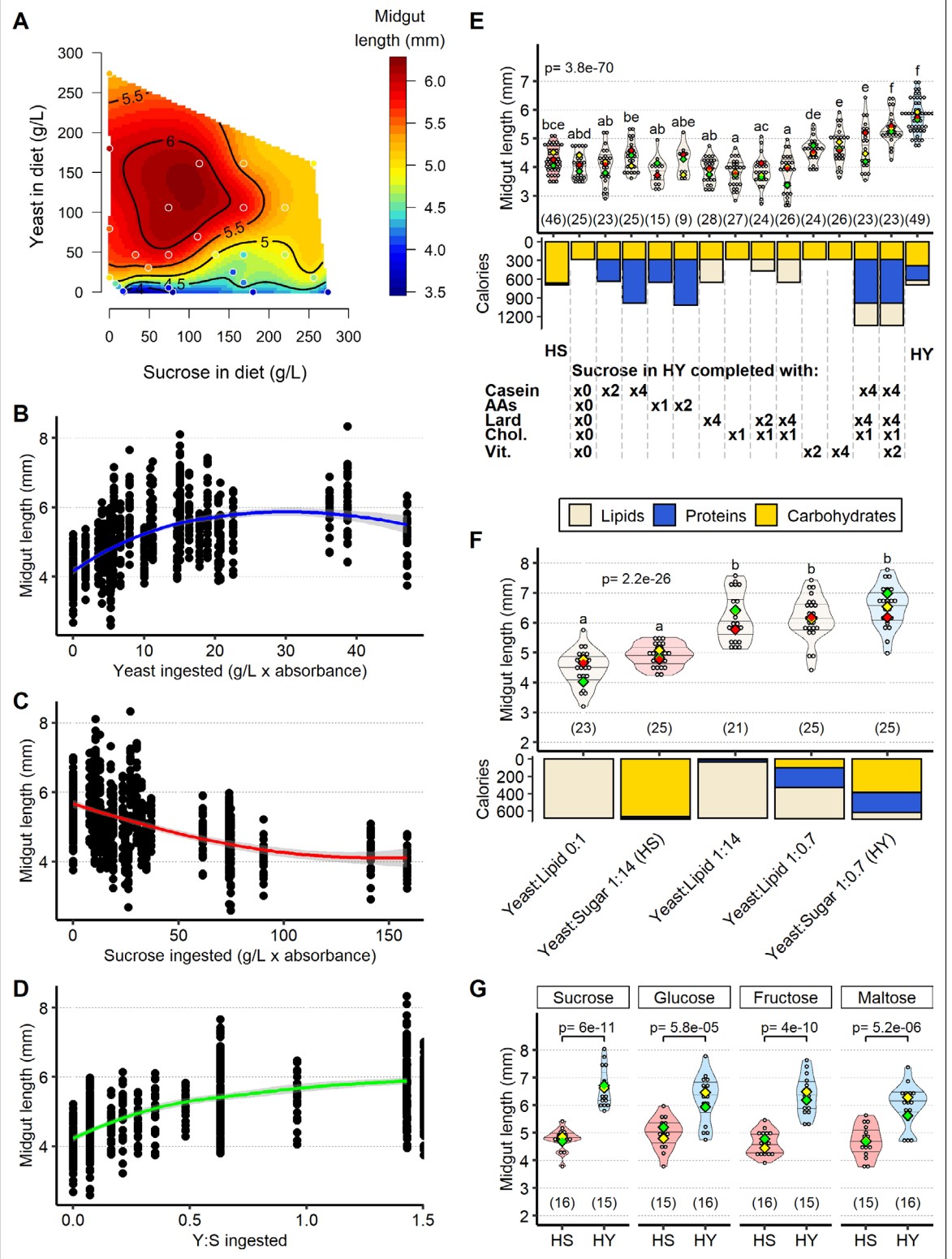

**Figure 2.** Sugar opposes yeast-induced increase of midgut length. (**A**) Midgut length is maximized at specific points in diet space. Adult flies were maintained for 5 days from eclosion on one of 28 diets based on different caloric concentration and yeast to sucrose ratios (see *Figure 2—figure supplement 1A* for scheme on diets used and sample size). The list of recipes can be found in *Supplementary file 1*. The figure shows contours of a thin-plate spline (Generalized Additive Model) of length (mm, coded by colors) as a function of yeast and sucrose in diet. Colored dots represent mean of samples in a particular diet. (**B–D**) Yeast and sucrose have mutually opposite impacts on midgut length. Plots show an increase in midgut length with

*Figure 2 continued on next page*

*Figure 2 continued*

increased amount of yeast ingested (**B**); a decrease in midgut length with increased amount of sucrose ingested (**C**); and an increase in midgut length with ratio of yeast to sucrose ingested (**D**). (**E**) Several nutrients from yeast (proteins, lipids, vitamins/minerals) are required to increase midgut length. Nutrients from yeast (proteins, amino acids, lipids, cholesterol, vitamins/minerals) were added against a base diet of only the amount of sucrose found in high yeast (HY) and devoid of yeast. Letters above violin plots represent grouping by statistical differences (post hoc Tukey on GLMM). Bars beneath the main plot describe caloric content provided by the different components. (**F**) Midgut size is opposed by sugar, but not other added calories. Diet with only lipids, isocaloric with high-sugar (HS) and HY diets, results in midguts of lengths comparable to those on HS diet. Substitution of sucrose from HS diet with isocaloric lipids (yeast:lipid 1:14) results in midguts as long as those on HY. Midguts of flies reared on a diet substituting sucrose in HY diet with lipids (yeast:lipid 1:0.7) are also similar in length to those of flies fed HY. Letters above violin plots represent grouping by statistical differences (post hoc Tukey on GLMM). Bottom part of the chart (bar graph) describes caloric content provided by the different components. (**G**) Opposition by sugar of yeast-induced growth is not specific to sucrose. Statistical comparisons were performed with HS vs. HY for each sugar. All flies for experiments in this figure were moved on the experimental diets at eclosion and dissected 5 days post eclosion. For the violin/dot plots shown in this figure, white dots represent single midgut measurements. Lozenges represent mean of repeats. Violin plots are color coded according to diets (HS = red, HY = light blue, cream for other diets). Numbers in parentheses at the bottom of charts indicate sample size. Additional information on the statistics can be found in *Supplementary file 2*.

The online version of this article includes the following figure supplement(s) for figure 2:

**Figure supplement 1.** Nutritional geometry reveals the influence of yeast and sugar on midgut length.

**Figure supplement 2.** Food texture is not responsible for size differences between high-sugar (HS) and high-yeast (HY) diets.

**Source data 1.** Numeric data for *Figure 2A–D*.

**Source data 2.** Numeric data for *Figure 2E*.

**Source data 3.** Numeric data for calories in *Figure 2E*.

**Source data 4.** Numeric data for *Figure 2F*.

**Source data 5.** Numeric data for calories in *Figure 2F*.

**Source data 6.** Numeric data for *Figure 2G*.

**Figure supplement 1—source data 1.** Numeric data for *Figure 2—figure supplement 1A*.

**Figure supplement 1—source data 2.** Numeric data for *Figure 2—figure supplement 1B*.

**Figure supplement 1—source data 3.** Numeric data for *Figure 2—figure supplement 1C*.

**Figure supplement 2—source data 1.** Numeric data for *Figure 2—figure supplement 2A*.

**Figure supplement 2—source data 2.** Numeric data for *Figure 2—figure supplement 2B*.

**Figure supplement 2—source data 3.** Numeric data for *Figure 2—figure supplement 2C*.

## Diet composition affects both cell number and EC size in the midgut

Organ size can originate both in changes in cell numbers and cell size (e.g., *Neufeld et al., 1998*). We asked whether diet composition influences cell number or cell size in the *Drosophila* midgut, with particular attention given to the region that responds the most to diet, the posterior midgut (region 4, *Figure 1G*; *Buchon et al., 2013*). By combining immunostaining and transgenic cell type-specific labels, we quantified the number of ISCs (esg$^+$ Su(H)$^-$, cells), EBs (esg$^+$, Su(H)$^+$ cells), EEs (Prospero$^+$ cells), and ECs (esg$^-$, prospero$^-$, larger polyploid cells) in the midguts of flies feeding on HS or HY diets (*Figure 3A-C*). All the different cell types increased in number on the HY diet compared to the HS diet (*Figure 3C*), and their relative proportions did not change (*Figure 3D*), demonstrating that posterior midguts resize without noticeable changes in cellular composition.

ECs are the biggest and most numerous cells in the *Drosophila* midgut. For this reason, we hypothesized that EC size could also contribute substantially to overall midgut size, and that their resizing was more likely to regulate organ size than equivalent resizing of other cell types. We therefore focused on EC size to evaluate whether changes in cell size could also underlie changes in midgut size between HS and HY. We used density of EC nuclei as a proxy for cell density and found that the density of ECs (ECs per μm$^2$ of midgut tissue) was lower on HY than on HS diet (*Figure 3A and B*, *Figure 3—figure supplement 1A*), suggesting a difference in cell size. Using junction markers (anti-mesh antibody), we labeled EC membranes, which allowed us to directly measure the surface of average EC cross-sections (as depicted in *Figure 3—figure supplement 1B*) as an indicator of EC size (*Figure 3E and F*). ECs were 154% larger (median size) on the HY diet than on the HS diet (*Figure 3G*). We also determined that EC height was increased on HY diet (*Figure 3—figure supplement 1C*), meaning the increased EC area was not due to lateral stretching but to an increase in total cellular volume. ECs are polyploid

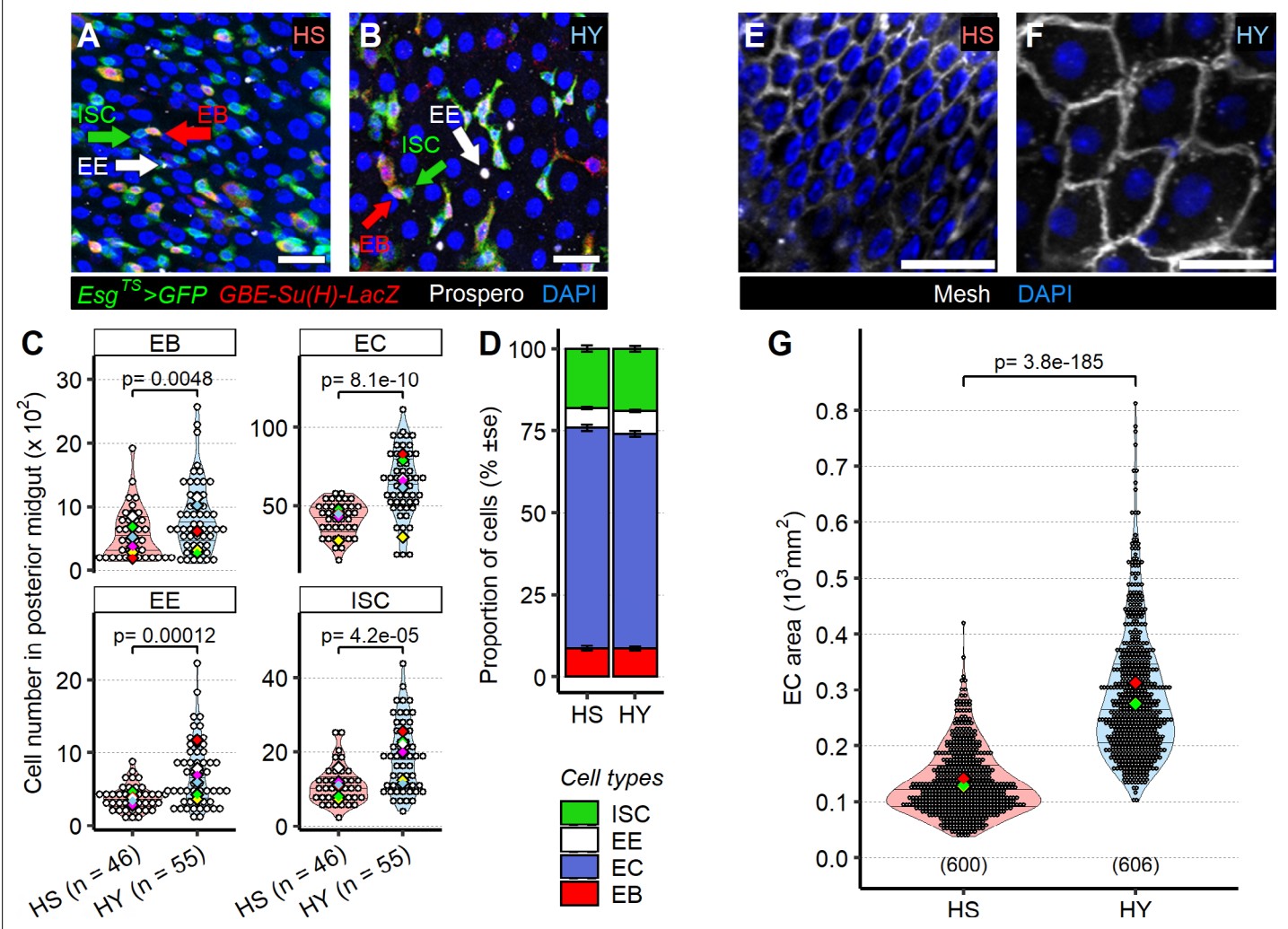

**Figure 3.** Diet composition affects both cell number and enterocyte size in the midgut. (**A–D**) High-yeast (HY) diet increases numbers of all midgut cell types. Representative pictures of midguts from flies kept on high-sugar (HS) (**A**) or HY (**B**) diet. Green arrows indicate intestinal stem cells (ISCs), marked only by GFP (green), red arrows indicate enteroblasts (EBs), marked by GFP and GBE Su(H)-lacZ (red), and white arrow indicate enteroendocrine (EE) cells, marked with anti-Prospero antibody (white). All nuclei are stained with DAPI (blue). Quantification of total cell numbers in the posterior midgut (R4) for HS and HY, statistical analysis is comparing HS vs. HY for each cell type (**C**). HS and HY diets do not affect the relative proportion of cell types in the midgut (error is standard error of the mean D). (**E–G**) Diet affects enterocyte size. Representative picture of midguts stained with anti-mesh antibody on HS (**E**) vs. HY (**F**) diet. Quantification of EC size of flies on HS or HY diet confirms an increase in cell size on HY diet (**G**). All flies for experiments in this figure were moved on the experimental diets at eclosion and dissected 5 days post eclosion. For the violin/dot plots shown in this figure, white dots represent single midgut measurements. Lozenges represent mean of repeats. Violin plots are color coded according to diets (HS = red, HY = light blue). Numbers in parentheses at the bottom of charts indicate sample size. Additional information on the statistics can be found in *Supplementary file 2*. Scale bars are 25 μm for all images.

The online version of this article includes the following figure supplement(s) for figure 3:

**Figure supplement 1.** Diet composition affects enterocyte (EC) size but not the cellular composition of the midgut epithelium.

**Source data 1.** Numeric data for *Figure 3C and D* and *Figure 3—figure supplement 1A*.

**Source data 2.** Numeric data for *Figure 3G*.

**Figure supplement 1—source data 1.** Numeric data for *Figure 3—figure supplement 1C*.

**Figure supplement 1—source data 2.** Numeric data for *Figure 3—figure supplement 1H*.

cells, and variation in ploidy can underlie variation in cell size (*Edgar and Orr-Weaver, 2001*; *Orr-Weaver, 2015*). Therefore, we asked if the increase in size of ECs was accompanied by an increase in ploidy. We dissociated nuclei and measured ploidy through FACS. Ploidy profiles were similar on HS and HY, suggesting that the difference in midgut size due to diet is not a result of a change in ploidy

(*Figure 3—figure supplement 1D–H*). Our results demonstrate that diet composition influences both the size and numbers of cells that build the *Drosophila* midgut.

## The midgut plastically resizes in response to shifts in diet composition

We next asked how the size difference between HS and HY is established, and if this is a plastic process. We measured the growth kinetics of midguts on the HS and HY diets during the first few days after emergence. The HY diet sustained continuous midgut growth in the first 5 days post eclosion, whereas length remained similar to that at eclosion on HS (*Figure 4—figure supplement 1A*). We next asked if, given enough time (up to 28 days), midguts on the HS diet would be able to grow to levels comparable to the HY diet. We found that midguts of flies kept on the HY diet increased in size, while the size of those kept on the HS diet decreased over the course of 28 days (*Figure 4—figure supplement 1B and C*). Does this change represent a developmental program triggered strictly post eclosion (*Buchon et al., 2013*) or a dynamic response to nutritional variation (*O'Brien et al., 2011*; *Obniski et al., 2018*)? We tested whether the midguts of flies maintained on either HS or HY can resize in response to subsequent dietary changes beyond the first 3 days of maturation (*Buchon et al., 2013*) using two different approaches. We first tested whether midguts of flies maintained on either HS or HY for 7, 14, or 21 days could still resize in response to a diet switch. Midguts still experienced HY-mediated increase in size after being on the HS diet for either 7, 14, or 21 days (*Figure 4—figure supplement 1D*), and midguts experienced HS-mediated shrinkage after being on the HY diet for 7, 14, or 21 days (*Figure 4—figure supplement 1E*), suggesting that a shift in diet composition can resize the midgut plastically during the fly's entire adult healthspan. We also switched flies alternately between HS and HY diets every 7 days for 3 weeks and found that their midgut was able to resize following multiple variations in diet (*Figure 4A*), reminiscent of what has been already documented for cell number plasticity in similar experiments (*O'Brien et al., 2011*). Altogether, this revealed that shifts in diet composition can plastically resize the midgut of *Drosophila*.

## Shifts in diet composition change absolute and relative levels of cell loss and ISC proliferation

We next asked whether changes in midgut size would be accompanied by changes in epithelial turnover. Study of environmental regulation of the midgut has so far focused largely on stem cell behavior. However, ISC proliferation is only half of the equation that governs epithelial turnover in the midgut. While most studies in *Drosophila* have focused exclusively on ISC proliferation and the associated 'cell gain,' the rate of ISC proliferation can only make sense of total cell gain when examined relative to the rate of cell loss. To understand diet-dependent midgut growth, we designed an integrative analysis of stem cell proliferation in the context of overall epithelial cell dynamics. We reasoned that any net change in cell number upon a diet shift must be the result of changes in absolute levels of cell gain (ISC proliferation) and cell loss, which together lead to a relative cell replacement ratio (i.e., the number of cells gained for each cell lost). Therefore, we measured each parameter on the HY and HS diets, and their response to diet switching.

We first quantified ISC proliferation in the midguts of flies fed HS or HY diets by immunostaining against the mitotic marker phospho-histone 3 (pH3). pH3-positive cells were more abundant on the HY diet compared to HS diet throughout the first 4 weeks of adulthood (*Figure 4B*). We also quantified this mitotic index across the diet space in which we had previously measured midgut length (*Figure 2A*). Indeed, ISC proliferation peaked in the same diets that were associated with long guts, showing promotion by yeast, but opposition by sugar (*Figure 4—figure supplement 1F and G*). We then asked whether ISC proliferation tracked the changes in length driven by switching between diets. Switching flies from HS to HY increased ISC proliferation (*Figure 4C*), and vice versa after switching from HY to HS (*Figure 4D*). Do these mitoses translate into more progeny (ECs and EEs)? We tested directly for an impact of diet on ISC proliferation by monitoring the number of new cells using the *esg*^F/O^ lineage tracing tool, which marks daughters of proliferating cells with GFP (*Jiang et al., 2009*). More cells were marked by GFP on the HY diet than on the HS diet (*Figure 4E and F*), indicating that HY-associated mitosis increases gain of new cells. These data indicate that diet composition alters ISC proliferation rate, with higher proliferation correlated to midgut size.

We then examined EC loss on the HS and HY diets. We reasoned that EC loss in the *Drosophila* midgut could be quantified by labeling ECs and EBs with a transient pulse of a long-term stable label

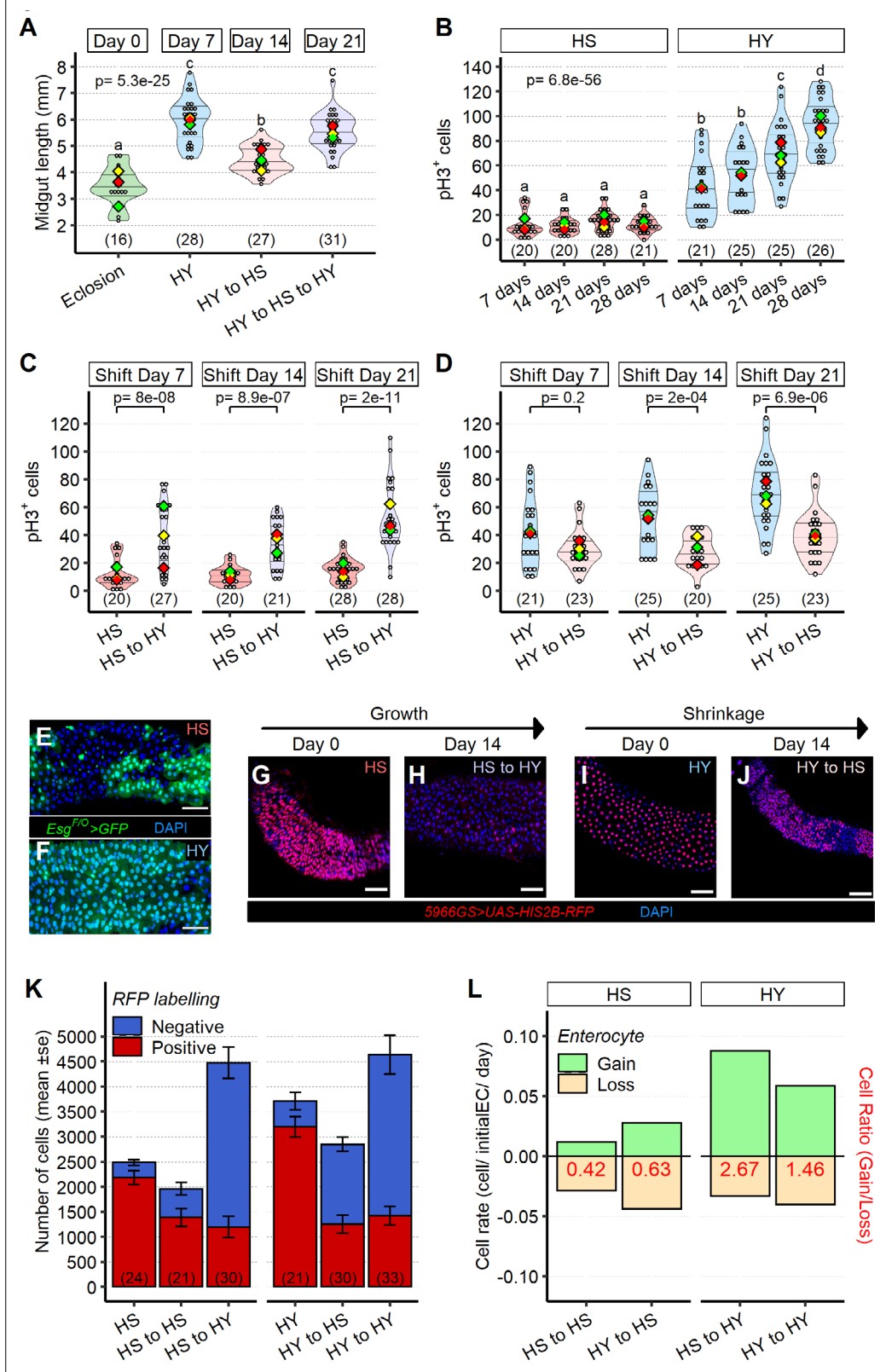

**Figure 4.** Shifts in diet composition lead to plastic midgut resizing and changes in absolute and relative cell loss and gain. (**A**) Midguts can respond plastically to changes in isocaloric diets. Midgut length increases from eclosion (day 0) on high yeast (HY) for 7 days, then decreases when switched to high sugar (HS) for additional 7 days but can reincrease size upon a further 7 days HY feeding. Letters above violin plots represent grouping by

*Figure 4 continued on next page*

*Figure 4 continued*

statistical differences (post hoc Tukey on GLMM). (**B**) Mitotically active cells visualized by phospho-histone H3 (pH3) immunostaining are more numerous on HY diet than on HS diet. pH3$^+$ cells gradually increase over time on HY, but not HS diet. Letters above violin plots represent grouping by statistical differences (post hoc Tukey on GLMM). Flies were put on the diets at eclosion, days on x-axis reflect days from eclosion. (**C, D**) Shifting between diets impacts pH3$^+$ cell number in both growth (HS to HY) and shrinkage (HY to HS) experiments. Days in chart are from eclosion. Statistical comparisons are vs. pre-shift measurement. (**E, F**) Clonal assay with *Esg$^{F/O}$* system put on either HS or HY from eclosion for 5 days, illustrates increased number of marked cells on HY (**F**) vs. HS (**E**) diets in region 4 of the midgut. GFP, in green, marks all cells made since the *Esg$^{F/O}$* system was activated. (**G–L**) Cell loss assay enables analysis of the impact of diet composition on replacement ratio and rate. Description of experimental design is found in Materials and methods and illustrated in *Figure 4—figure supplement 1H*. In brief, this assay allows us to mark enterocytes (ECs) and enteroblasts (EBs) at the start of the experiment (9 days post eclosion) and to count their numbers 14 days after shifting dietary conditions (23 days post eclosion) recapitulating growth and shrinkage of the midgut, thus estimating cell gain and cell loss in these conditions. Representative pictures for the cell loss assay in growing conditions (**G, H**) and shrinkage conditions (**I, J**). In red *5966GS>His2B-RFP*, marking EB and EC. Number of ECs in the posterior midgut, both marked (red, old ECs) and unmarked (blue, new ECs) by RFP, error bars are SE from three repeats (**K**). Data shown as rate relative to experiment start (cell/initial EC/day) (**L**). Number on bar in red is the ratio of EC gained/EC lost (see Materials and methods for formula). For the violin/ dot plots shown in this figure, white dots represent single midgut measurements. Lozenges represent means of replicate experiments. Violin plots are color coded according to diets (green = eclosion, HS = red, HY = light blue, HY to HS = pink, HS to HY = purple). Numbers in parentheses at the bottom of charts indicate sample sizes. Additional information on statistics can be found in *Supplementary file 2*. Scale bars are 50 µm for all images.

The online version of this article includes the following figure supplement(s) for figure 4:

**Figure supplement 1.** Shifts in diet composition lead to midgut resizing and are associated with changes in the absolute and relative rates of cell loss and gain.

**Figure supplement 2.** Illustration and quality control for cell loss assay.

**Source data 1.** Numeric data for *Figure 4A*.

**Figure supplement 1—source data 1.** Numeric data for *Figure 4B* and *Figure 4—figure supplement 1C*.

**Figure supplement 1—source data 2.** Numeric data for *Figure 4C* and *Figure 4—figure supplement 1D*.

**Figure supplement 1—source data 3.** Numeric data for *Figure 4D* and *Figure 4—figure supplement 1E*.

**Source data 2.** Numeric data for *Figure 4K*.

**Source data 3.** Numeric data for *Figure 4L* and *Figure 4—figure supplement 2D*.

**Figure supplement 1—source data 4.** Numeric data for *Figure 4—figure supplement 1A*.

**Figure supplement 1—source data 5.** Numeric data for *Figure 4—figure supplement 1F and G*.

**Figure supplement 2—source data 1.** Numeric data for *Figure 4—figure supplement 2B*.

**Figure supplement 2—source data 2.** Numeric data for *Figure 4—figure supplement 2C*.

(*Figure 4—figure supplement 2A*). Conducting this pulse chase over 14 days, which is a period longer than that in which we had recorded diet-induced midgut resizing, would allow us to quantify cell loss accompanying size plasticity. Histone2B-RFP (His2B-RFP) labels nuclei extremely stably (*Antonello et al., 2015*). We confirmed that His2B-RFP was not quenched up to 14 days after transient expression in tissues that do not turnover cells in the same manner as the midgut (crop and hindgut), in both HS and HY conditions (*Figure 4—figure supplement 2B*). This result indicates that, in ECs, a loss of RFP$^+$ nuclei would reflect a loss of cells, not a loss of fluorophore. We expressed His2B-RFP transiently (using the hormone-dependent *5966GS* EB-EC-specific driver and 3 days of feeding the inducer, RU486) (*Figure 4—figure supplement 2A*), after which RFP was detectable in most ECs (*Figure 4G,I*). Five days after start of the chase, no cell loss was detectable at the organ level in midguts maintained constantly on HY diet (HY to HY condition, *Figure 4—figure supplement 2C*), so we proceeded to quantify a later timepoint (14 days post chase start, *Figure 4K*). By 14 days after chase start, some, but not all, ECs retained RFP fluorescence (*Figure 4H and J*). This system therefore allowed us to quantify the cells gained (increase in DAPI$^+$ RFP$^-$ cells) and lost (decrease in DAPI$^+$ RFP$^+$ cells) on each of the HS and HY diets (*Figure 4G–K*, *Figure 4—figure supplement 2A*). This experiment showed that numerous cells are gained on the HY diet, independent of previous diet (i.e., HS to HY and HY to HY), but also that a high number of cells are lost when flies started the experiment on

the HY diet, independent of the present diet (i.e., HY to HS and HY to HY, *Figure 4—figure supplement 2D*), suggesting that current diet and dietary history could both influence tissue turnover. We note that, as guts on the HY diet are bigger, dietary history could just reflect initial midgut size. To better characterize the coupling between cell gain and cell loss, we further calculated the rates of cell gain and loss per initial EC per day (number of cells gained/lost per initial EC per unit of time, *Figure 4L*) as this rate takes into account size difference. Surprisingly, the rate of EC loss (per initial EC) was higher on the HY diet (HY to HY) than on the HS diet (HS to HS). This suggested that, taken alone, the mere loss of ECs cannot fully explain the small size of guts on the HS diet.

What was the balance of relative cell gain and loss upon HY or HS feeding? We compared the results of our ISC proliferation and EC loss indices (*Figure 4K and L*, *Figure 4—figure supplement 2D*). When midguts were growing on HY, cell gain exceeded loss, generating an overall replacement ratio greater than 1 (*Figure 4L*). Accordingly, when midguts were shrinking on HS, the replacement ratio was lower than 1 (*Figure 4L*). Of note, in none of our four conditions we did detect a replacement ratio close to 1, indicating that neither HS nor HY diets supported a strict coupling between cell gain and loss. We note that, importantly, similar replacement ratios occurred with very different absolute rates of cell gain and loss. For instance, replacement ratio was similar when flies were constantly fed the HS diet, or after switching from an HY to an HS diet (replacement ratio ~0.5), despite considerably different levels of absolute cell gain and loss in these two different conditions. In addition, the replacement ratio was higher in flies switched from HS to HY diet than in flies constantly feeding on the HY diet, confirming an influence of dietary history on the rates of cell gain and loss. This indicates that neither ISC proliferation nor cell loss alone capture the nature of epithelial turnover; rather, quantification of both parameters and calculation of the replacement ratio is required. This also indicates that the diet on which flies are feeding on is not the only factor influencing the absolute and relative rates of cell gain and loss, the dietary history or possibly initial organ size are also key factors. However, additional factors, in addition to cell proliferation and cell loss, must control the size of the midgut since the cell replacement ratio does not entirely translate into the increase in midgut size (e.g., on HY diet, replacement ratio is 1.46 while midgut area posterior increase is 1.14). Altogether, our data demonstrate that diet composition independently alters the rates of cell gain and cell loss, and their relative ratio underpins the emergent property of midgut growth.

## Sugar induces translational stress in the midgut, which uncouples ISC proliferation from expression of pro-mitotic niche signals

Our phenomenological investigations revealed several unexpected results: (1) yeast and sugar have opposing effects on midgut size; (2) organ size is determined by both cell number and cell size; and (3) epithelial cell gain and loss, induced by dietary variation, do not linearly follow patterns intuitively expected based on midgut size. We wondered what mechanistic processes might underlie these observations. We performed an RNA-seq analysis of dissected midguts of flies that were fed either the HS or HY diet from eclosion and examined the kinetics of gene expression from eclosion on to days 1, 2, 3, and 5. An unsupervised method of grouping samples according to gene expression (principal components analysis) revealed a very clear two-phase process (*Figure 5A*). An apparently programmatic series of changes in gene expression occurred during the first day. These changes were largely diet-independent and correspond to maturation of the midgut (*Buchon et al., 2013*). However, in the following days, transcriptomes diverged substantially between the HS and HY diets. We identified differentially expressed genes (DEGs) between the HS and HY diets at days 3 and 5. Gene ontology enrichment analysis of the genes significantly more expressed on one diet or the other revealed striking differences (*Figure 5B*). Genes upregulated on the HY diet included numerous genes involved in digestion (proteolysis, carbohydrate metabolism, lipid metabolism, sterol transport) and respiration or oxidative stress (examples in *Figure 5C*). Among the digestive enzymes identified as upregulated on the HY diet, proteases, peptidases, and amino acid metabolic processing enzymes were 10–20-fold upregulated relative to HS, corresponding to dietary protein content. Similar results were found for genes involved in sterol transport, lipid metabolism, and carbohydrate digestion. These results collectively suggest higher digestion of protein, lipids, and carbohydrates on the HY diet, even though HS contained more sugar. On the HS diet, unexpectedly, genes upregulated compared to on the HY diet were ones involved in the regulation of growth, tissue development, proliferation, response to stress, and signal transduction (*Figure 5B*), despite the HS diet being growth repressive. Together, these

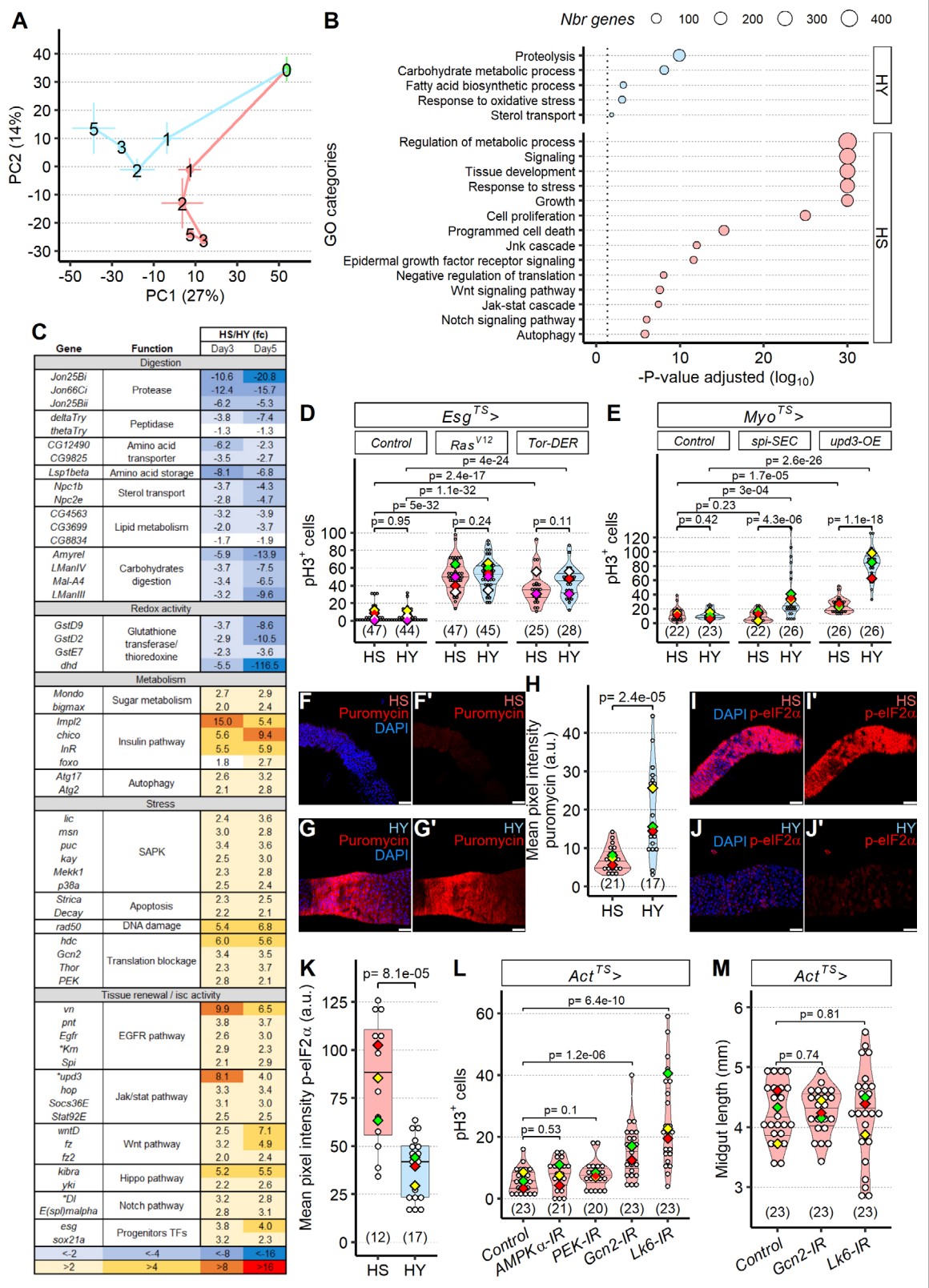

**Figure 5.** Sugar uncouples intestinal stem cell (ISC) proliferation from niche signal expression by inducing translational stress. (**A**) Diet influences midgut transcriptomes after an initial programmed developmental transition. The plot shows a principal components analysis (PCA) of the whole transcriptome, with means per diet per day ± standard error (three repeats). Numbers on the plot represent the day of dissection from eclosion. Lines connect the datapoints sequentially (day 0 to day 1, day 1 to day 2, etc.), and show the divergent transcriptomic trajectory followed by midguts on

*Figure 5 continued on next page*

*Figure 5 continued*

the two different diets from eclosion. (**B, C**) Diet modulates expression of functionally distinct gene classes. Midguts of flies fed high-yeast (HY) diet show higher expression of genes with digestive functions, while high-sugar (HS) diet involves mainly genes attributed to stress response and growth. X-axis represents the statistical significance of the gene ontology (GO) categories (y-axis) after adjustment for multiple testing. Size of the dot is proportional to number of genes in the given GO category (**B**). Table of genes significantly differently expressed, between HS and HY, as a ratio of HS/HY, representing midgut response to HS and HY diets; additional information on the statistics is found in Materials and methods; asterisks denote genes significantly different for p-value but not for adjusted p-value (**C**). (**D**) Cell proliferation is possible on HS diet when genetically induced. Progenitor-specific (*Esg^TS*) overexpression of a constitutively active form of Ras (*UAS-Ras^V12*) and of *UAS-Tor-DER* (EGFR Active), both known proliferative inducers, allows for increased proliferation on HS diet. Flies were 6 days old when dissected. (**E**) Enterocyte-specific overexpression (*Myo^TS*) of *UAS-upd3-OE* and *UAS-spi-SEC* elicit increased proliferation, strongly only on HY diet, and weakly on HS with *UAS-upd3-OE*. Flies were 9 days old when dissected. (**F–H**) General translation is lower on HS than HY, shown by puromycin incorporation assay. Images show lower incorporation on HS (**F, F'**) than HY (**G, G'**) in region 4 of the midgut from 5 -day-old Canton-S (Cs) flies that were shifted on HS or HY at eclosion. Quantification of mean pixel intensity of puromycin stain (**H**). (**I–K**) p-eIF2α stain is elevated on HS (**I, I'**) compared to HY (**J, J'**). Quantification of mean pixel intensity of p-eIF2α stain in region 4 of the midgut from 5 -day-old Cs flies that were shifted on HS or HY at eclosion (**K**). (**L**) Re-enabling translation can restore mitosis in midguts shrinking after being shifted from HY to HS diet for 7 days. Initial shift was performed 12 days post eclosion. Blocking translational inhibition with *Act^TS>Gcn2 IR* or *Act^TS>LK6 IR* is sufficient to increase pH3+ cells in midguts of flies on HS diet. However, *Act^TS>PEK IR* and *Act^TS>AMPKα-IR* had no effect on the number of pH3+ cells. Statistical comparisons are vs. control. (**M**) Despite increased mitotic activity following repression of translational inhibition in *Act^TS>Gcn2 IR* or *Act^TS>LK6* IR, midgut size was still reduced on flies kept on HS diet. The statistical comparison is comparing interaction between diet and fly lines. For the violin/dot plots and boxplots showed in this figure, white dots represent single midgut measurements. Lozenges represent mean of repeats. PCA, violin plots, box plots, and the PCA are color coded according to diets (green = eclosion, HS = red, HY = light blue). Numbers in parentheses at the bottom of charts indicate sample size. Additional information on the statistics can be found in *Supplementary file 2*. Scale bars are 50 μm for all images.

The online version of this article includes the following figure supplement(s) for figure 5:

**Figure supplement 1.** High-sugar (HS) diet is associated with shorter lifespan.

**Figure supplement 2.** Translational stress is upregulated in the epithelium on high-sugar (HS) diet.

**Source data 1.** RNA-seq count data for *Figure 5A–C*.

**Source data 2.** Selected gene ontology (GO) categories for *Figure 5B*.

**Source data 3.** All gene ontology (GO) results for *Figure 5B*.

**Source data 4.** Differentially expressed genes for high sugar (HS) vs. high yeast (HY) (days 3 and 5), from which representative genes are shown in *Figure 5C*.

**Source data 5.** Numeric data for *Figure 5D*.

**Source data 6.** Numeric data for *Figure 5E*.

**Source data 7.** Numeric data for *Figure 5H*.

**Source data 8.** Numeric data for *Figure 5K*.

**Source data 9.** Numeric data for *Figure 5L*.

**Source data 10.** Numeric data for *Figure 5M*.

**Figure supplement 1—source data 1.** Numeric data for *Figure 5—figure supplement 1A*.

**Figure supplement 2—source data 1.** Numeric data for *Figure 5—figure supplement 2E*.

**Figure supplement 2—source data 2.** Numeric data for *Figure 5—figure supplement 2F*.

**Figure supplement 2—source data 3.** Numeric data for *Figure 5—figure supplement 2G*.

findings suggest conflicting processes of cell proliferation/growth versus arrest, in which growth-promoting mechanisms are activated by, but fail to overcome, the effect of sugar.

We examined the pro-proliferative genes more highly expressed on the HS diet in detail and found many genes shown in previous studies to be linked to epithelial stress and compensatory turnover (*Buchon et al., 2009b*). These included genes involved in SAPK signaling (JNK, P38), apoptosis, as well as most pathways controlling ISC proliferation and differentiation (EGFR, JAK-STAT Wnt, hippo, Notch pathways). Importantly, genes coding for classic pro-proliferative niche signals (*vn, Krn, Spi, wntD, upd3*) were also more highly expressed. To date, these signals have been shown to be upregulated in the niche in response to stress, providing a cue for ISCs to proliferate and replace damaged or lost differentiated cells (*Bonfini et al., 2016*). Accordingly, we detected a shorter lifespan on HS diet compared to HY diet, confirming the deleterious impact of the HS diet (*Figure 5—figure supplement 1A*). Upregulation of these pathways in response to the HS diet suggested that the midguts on HS diet are stressed, yet ISCs failed to respond (*Figure 4*). This raised the question of whether ISCs on

the HS diet are unable to proliferate at all, or whether HS uncouples nonautonomous niche signaling from an appropriate proliferative response. Such uncoupling could result from the inability of ISCs to respond to niche-derived signals or from the inability of the niche to translate or secrete niche signals, leading to an indirect decrease in proliferation.

We therefore first asked whether ISCs of flies fed the HS diet had the ability to proliferate in response to a pro-mitotic cell-autonomous signal. In progenitor cells, we overexpressed the oncogene $Ras^{V12}$ ($esg^{TS}>UAS-Ras^{V12}$), a driver of cell proliferation, and $UAS-Tor-DER$ (an activator of EGFR pathway) and again quantified pH3-positive cells after feeding on HS or HY diet. With overexpression of both constructs, ISC proliferation was higher than in control flies and statistically indistinguishable in the same genotype between HS and HY diets (*Figure 5D*). These results confirm that ISCs retain proliferative capacity on the HS diet, but proliferation is uncoupled from expression of pro-mitotic genes. Having shown cell-autonomous proliferative capacity was retained, we then asked whether HS diet uncouples ISC proliferation from non-autonomous niche signals by testing if stem cell proliferation could be induced through over expression of two ligands from the niche (ECs), *upd3* and *spitz*, which are known to be involved in the response to stress in the midgut (*Buchon et al., 2010*). We used $Myo^{TS}$ to drive overexpression of $UAS-upd3-OE$ and $UAS-spi-SEC$ in ISCs. Both constructs were able to significantly drive proliferation on HY diet; however, on the HS diet *spitz* overexpression did not lead to increase proliferation and overexpression of *upd3* led to a smaller degree of increase in proliferation. This result suggests that diet uncouples niche signals from stem cell activity by acting on the niche itself or blocking ISCs from receiving said signals (*Figure 5E*).

We next asked how ISC proliferation could become uncoupled from the high expression of niche derived pro-proliferative signals. Our RNAseq experiment indicated that the midguts of flies feeding on the HS diet express higher levels of genes involved in translation inhibition, including *PEK* and the *Gcn2* kinase (*Figure 5B and C*). Reduced translation of niche-derived signals has been previously shown to impair midgut repair after infection (*Chakrabarti et al., 2012*), and we hypothesized a similar effect could occur upon HS feeding. We first assessed whether global translation was lower in midguts of flies feeding on HS than on HY by measuring puromycin incorporation 3 hr after feeding. Puromycin incorporation was significantly higher on HY than HS, demonstrating that general translation was lower on HS diet (*Figure 5F–H*). A central regulator of global translation is eIF2α, which mediates a decrease in global translation when phosphorylated during metabolic stress. In our RNAseq, expression of two eIF2α kinases (*PEK* and *Gcn2*) was increased on the HS diet, suggesting that diet-dependent eIF2α phosphorylation could explain ISC-niche uncoupling. Immunostaining confirmed that eIF2α phosphorylation was higher on the HS diet (*Figure 5I–K*). p-eIF2α was detected both in progenitor cells and ECs but absent or only slightly detected in the visceral muscles (*Figure 5—figure supplement 2A, D'*). To directly test the role of translational inhibition in sugar-induced blockage of ISC proliferation, we knocked down expression of multiple genes involved in translation inhibition, including *Gcn2*, *Lk6*, *AMPKα,* and *PEK*, under control of an inducible and ubiquitous *Actin* that could act in all the midgut cell types that displayed eIF2α phosphorylation. Knockdown with RNAi of either *Gcn2* (an eIF2α kinase) or *Lk6* (an eIF4E1 kinase) in adult flies increased ISC proliferation on the HS diet (*Figure 5L*), or when shifting for a week from HY to HS diet (*Figure 5—figure supplement 2E*), compared to controls. However, global expression of the same RNAi constructs was not able to increase midgut length on the HS diet (*Figure 5M*) or to prevent midgut shrinking when transitioning from HY to HS (*Figure 5—figure supplement 2F*). To determine in which cell type translational inhibition had a role in blocking stem cell proliferation, we knocked down *Gcn2* in progenitor cells ($Esg^{TS}$) and ECs ($Myo^{TS}$). We found that it was possible to increase proliferation when knocking *Gcn2* in ECs, but not in progenitors (*Figure 5—figure supplement 2G*). This, in addition with previous results (*Figure 5D and E*), suggests that on the HS diet a translational inhibition in ECs involving Gcn2 uncouples ISC proliferation from increased expression of pro-proliferative niche signals.

## The midgut can resize independently of stem cell proliferation

To date, ISC proliferation has been assumed to underpin midgut growth in response to nutrition (*O'Brien et al., 2011*). However, our finding that restoring ISC proliferation was not sufficient to increase size on HS diet or abrogate the shrinking effect of switching from HY to HS diet indicated that additional mechanisms must underlie midgut size plasticity. We first explored this using natural population variation in the DGRP. We focused on a panel of seven DGRP lines (*Figure 6A*) including

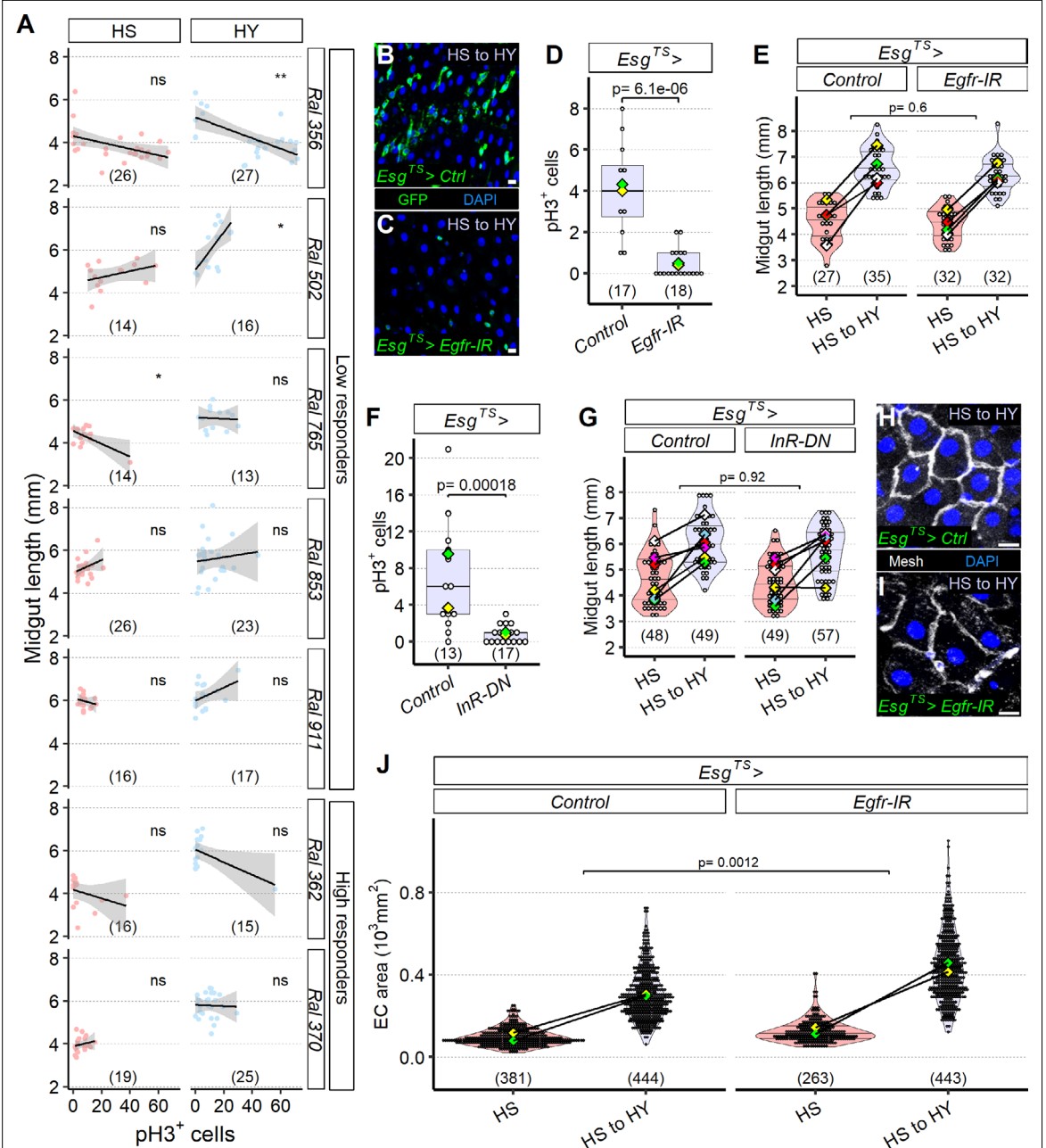

**Figure 6.** The midgut can resize independently of stem cell proliferation. (**A**) Mitotic index does not correlate with midgut length. Quantification of pH3+ cells across a selected panel of high and low responder *Drosophila* Genetic Reference Panel (DGRP) lines shows that midgut length does not correlate with cell proliferation, flies were moved on the experimental diets at eclosion and dissected 5 days post eclosion. (**B–E**) Blocking EGF signaling with *UAS-Egfr-IR* in progenitor cells (**C**) results in a change in conformation of progenitor cells, which assume a more rounded shape compared to the traditional triangular shape and seem to be less in number (not quantified) compared to control (**B**), consistent with lower pH3+ counts than control (**D**). Statistical comparison for (**D**) is for the interaction between diet and genotype. Data for (**D**) is from 19 days post eclosion. However, these *Esg^TS^>UAS-Egfr-IR* midguts are still able to reach a similar length to controls (**E**). In (**E**), high sugar (HS) was dissected at 12 days post eclosion, and HS to high yeast (HY) 7 days later. (**F, G**) Insulin signaling with a dominant negative construct in progenitor cells results in less proliferation (**F**) despite *Esg^TS^>UAS-InR-DN* resulting in the same midgut length growth as the control (**G**). Statistical comparison for (**G**) is for the interaction between diet and genotype. Data for (**F**) is from 19 days post eclosion. In (**G**), HS was dissected at 12 days post eclosion, and HS to HY 7 days later. (**H–J**) Increase in midgut length despite proliferation blockage is accompanied with compensatory area increase of enterocyte (EC). Representative pictures of midguts stained with membrane marker mesh (white), shifted from HS to HY at 12 days post eclosion, and kept on HY for seven additional days show bigger cells on *Esg^TS^>UAS-Egfr-IR* (**I**) compared to control (**H**). Quantification of EC cell size shows compensatory effect in ECs (**J**). Statistical comparisons for (**E, G, J**) are for the interaction between diet and genotype. For the violin/dot plots and boxplots showed in this figure, white dots represent single midgut measurements. Lozenges

*Figure 6 continued on next page*

*Figure 6 continued*

represent mean of repeats. Violin plots and box plots are color coded according to diets (HS = red, HY = light blue, HS to HY = purple). Black lines connecting means visualize the interaction between diets. Numbers in parentheses at the bottom of charts indicate sample sizes. Additional information on the statistics can be found in *Supplementary file 2*. Scale bars are 10 μm for all images.

The online version of this article includes the following figure supplement(s) for figure 6:

**Figure supplement 1.** Diet-dependent changes in mitotic index do not correlate with changes in midgut size.

**Source data 1.** Numeric data for *Figure 6A*, *Figure 6—figure supplement 1A*, *Figure 7A*, and *Figure 7—figure supplement 1B*.

**Source data 2.** Numeric data for *Figure 6D*.

**Source data 3.** Numeric data for *Figure 6E*.

**Source data 4.** Numeric data for *Figure 6F*.

**Source data 5.** Numeric data for *Figure 6G*.

**Source data 6.** Numeric data for *Figure 6J*.

**Figure supplement 1—source data 1.** Numeric data for *Figure 6—figure supplement 1F*.

lines that respond and lines that do not respond to diet composition, but have either a short midgut, a middle-sized midgut, and a long midgut; we measured midgut length and ISC proliferation rate (pH3 immunostaining). We first noticed, as previously published, that the proliferative response was variable across different *Drosophila* lines (*Figure 6A*; *Tamamouna et al., 2020*). ISC proliferation and midgut size were not positively correlated amongst these genotypes on either diet, except for one line (*Figure 6A*). In addition, midgut resizing did not correlate with change in proliferation between the HS and HY diets (*Figure 6—figure supplement 1A*), suggesting that proliferation could be dispensable for resizing.

We next directly tested whether ISCs are required for the midgut to resize. EGFR signaling is central to ISC function (*Buchon et al., 2010*; *Jin et al., 2015*). We confirmed that RNAi knockdown of the EGF receptor (*esg^TS^>Egfr* IR) greatly reduced the number of progenitor cells and resulted in a complete loss of proliferative cells (*Figure 6B–D*). However, despite the loss of proliferative cells, these flies were still able to grow their midgut upon switching from HS to HY diets (*Figure 6E*). In addition to the EGFR pathway, insulin/IGF-like signaling is a master regulator of ISC proliferation, notably in response to diet (*Amcheslavsky et al., 2009*; *Biteau et al., 2010*; *Choi et al., 2011*; *O'Brien et al., 2011*; *Strilbytska et al., 2020*; *Veenstra et al., 2008*). We therefore repressed the insulin pathway by overexpressing a dominant-negative insulin receptor in progenitor cells (*esg^TS^>InR* DN). This strongly decreased ISC proliferation (*Figure 6F*), but these midguts were nevertheless able to grow on the HY diet (*Figure 6G*). We observed similar results when we overexpressed the pro-apoptotic gene *reaper* in progenitor cells (*esg^TS^>rpr OE*), which resulted in complete loss of marked progenitor cells in region 4 of midguts already at 7 days post TARGET system activation on the HS diet (*Figure 6—figure supplement 1C*) compared to control midguts (*Figure 6—figure supplement 1B*), loss which was also evident after shifting flies on HY from HS for an additional week (*Figure 6—figure supplement 1E*). However, midguts were still able to increase in size similarly to controls (*Figure 6—figure supplement 1F*). Altogether, these results demonstrate that, surprisingly, proliferative capacity in ISCs can be dispensable for diet-dependent midgut resizing.

Our original hypothetical model was that midgut size would be regulated by the sum of cell gain, cell loss, and cell size, according to the morphological changes observed in *Figure 3*. While long-term maintenance of a stem cell pool is presumably important for survival, our present functional genetic data show that, in the short term, midgut resizing does not strictly require ISC proliferation. How? We noticed that ECs were larger when EGFR signaling was reduced in progenitor cells (*Figure 6H and I*). When we quantified EC sectional area in *esg^TS^>EGFR* IR flies, we found that it did indeed increase after switching from HS to HY diets, and more than in control midguts (*Figure 6J*). We visualized a similar morphology also in *esg^TS^>rpr* OE midguts shifted from HS to HY (*Figure 6—figure supplement 1E*). This suggests that EC size can compensate for the lack of increase in cell number when ISC proliferation is blocked, outlining a possible multicellular homeostatic mechanism involving a feedback between ISCs and ECs.

## EC resizing is necessary for diet-dependent midgut plasticity

Our data suggested that EC size could be an essential driver of midgut resizing. We first asked if EC size covaries with midgut size, as measured in *Figure 4A*. Indeed, EC size corresponded to midgut size across diet manipulations (*Figure 7—figure supplement 1A*). We wondered whether this correspondence would generalize across genetically variable genotypes. We went back to the seven DGRP lines we previously analyzed for proliferation and length and, in the same midguts, measured EC size (*Figure 7A*). EC size and midgut length were generally positively correlated, sometimes very strongly, especially on HY diet. Additionally, midgut resizing correlated with change in EC area between the HS and HY diets (*Figure 7—figure supplement 1B*), suggesting that EC size could be a strong driver of midgut plasticity.

We attempted to manipulate EC size by functional genetics to test the role of EC size on midgut resizing. We focused on the TOR pathway because of its evolutionarily conserved role as a regulator of cell size (*Blenis, 2017*; *Gonzalez and Rallis, 2017*). Additionally, we found in our RNA-seq analysis that several targets of Foxo, including *foxo* itself, are upregulated on HS diet (*chico*, *InR*, *Impl2*, *Thor*, *Figure 5C*), indicating that the TOR pathway is downregulated on HS diet (*Hay, 2011*; *Rera et al., 2012*). We also separately tested a reported of Foxo activity, *thor-lacZ* (*Karpac et al., 2011*), and found that it upregulated on HS diet compared to HY (*Figure 7—figure supplement 2A, C*). This confirmed that physiological TOR signaling is modulated by diet. We first asked whether the TOR pathway could control EC size cell autonomously using a clone-tracing system (*hsFlp; Act>STOP >Gal4,UAS-GFP*), which allowed us to selectively inhibit (*UAS-Tor-IR*) or activate (*UAS-Rheb-OE*) the TOR pathway specifically in fluorescently labeled single ECs. TOR activation enlarged ECs, and knock-down decreased EC size (*Figure 7B–D*), confirming that TOR is a cell-autonomous regulator of EC size. We next asked whether TOR was required to modify EC size in response to diet. When TOR was knocked down in the ECs (*Myo^{TS}>UAS-TOR-IR*, *Figure 7E–G*) of midguts switched from HS to HY, change in EC size was attenuated, confirming that the TOR pathway mediates diet-dependent EC resizing.

We next asked whether TOR pathway's effect on diet-dependent EC resizing was sufficient to alter diet's effect on overall midgut size. We knocked down a range of genes coding for proteins in the TOR pathway and measured midgut growth after switching from HS to HY. Each of these knockdowns qualitatively diminished growth, and this effect was statistically significant in most cases (*Figure 7H*). Knockdown of *Tor* itself (*Myo^{TS}>UAS-TOR-IR*) led to a significant decrease in growth compared to control when moved on the HY diet, nearly abolishing any change in size. Furthermore, repeating this experiment using a gene-switch system (i.e., *5966GS*) to drive RNAi against Tor in ECs led to similar results (*Figure 7—figure supplement 2D*). We then tested the impact of other components of TOR signaling. TOR is found in two macromolecular complexes, mTORC1 and 2, respectively (*Wullschleger et al., 2006*). The mTORC1 complex integrates Raptor, and EC-specific RNAi against Raptor was not statistically significant, but qualitatively diminished regrowth after switching from HS to HY (*Figure 7H*). Other classical components of the TOR pathway, including the transcription factors Myc and SREBP, as well as the kinase S6K, were all significantly required for regrowth (*Figure 7H*). Together, these data demonstrate that the TOR pathway regulates the growth of ECs, and that EC resizing is required for concomitant increase in organ size. mTORC1 is a negative regulator of autophagy (*Dossou and Basu, 2019*; *Kim and Guan, 2019*). In addition, our RNA-seq experiment (*Figure 5*) showed that genes related to autophagy were significantly upregulated on HS diet (*Atg2*, *Atg17*, *Atg101*). These observations outlined a hypothesis in which TOR repression on the HS diet increases autophagy, consequently decreasing cell and midgut size. *Atg2* and *Atg8a* are core essential genes required for macro-autophagy (required for phagophore and autophagosome formation) and downstream of TOR pathway (*Mulakkal et al., 2014*). We first confirmed that autophagy could influence EC size cell-autonomously by knocking down *Atg2* with a clone-tracing system. RNAi of *Atg2* in singly labeled ECs (*hsFlp; Act>STOP >Gal4, UAS-GFP>UAS-Atg2-IR*) produced larger cells, consistent with our hypothesis that autophagy reduces EC size (*Figure 7I and J*). We then tested the requirement for *Atg2* and *Atg8a* for dietary regulation of midgut size. We knocked down *Atg8a* (*Myo^{TS}>UAS-Atg8a-IR*) or *Atg2* (*Myo^{TS}>UAS-Atg2-IR*) in ECs specifically and measured the impact on midgut shrinkage when flies transition from the HY to the HS diet (*Figure 7K and L*). Each Atg gene was required for complete shrinkage after switching from HY to HS diets. Performing this experiment with a gene-switch system (i.e., *5966GS*) to drive RNAi led to similar results (*Figure 7—figure supplement 2E*). Altogether, our results demonstrate that the TOR/autophagy pathway is

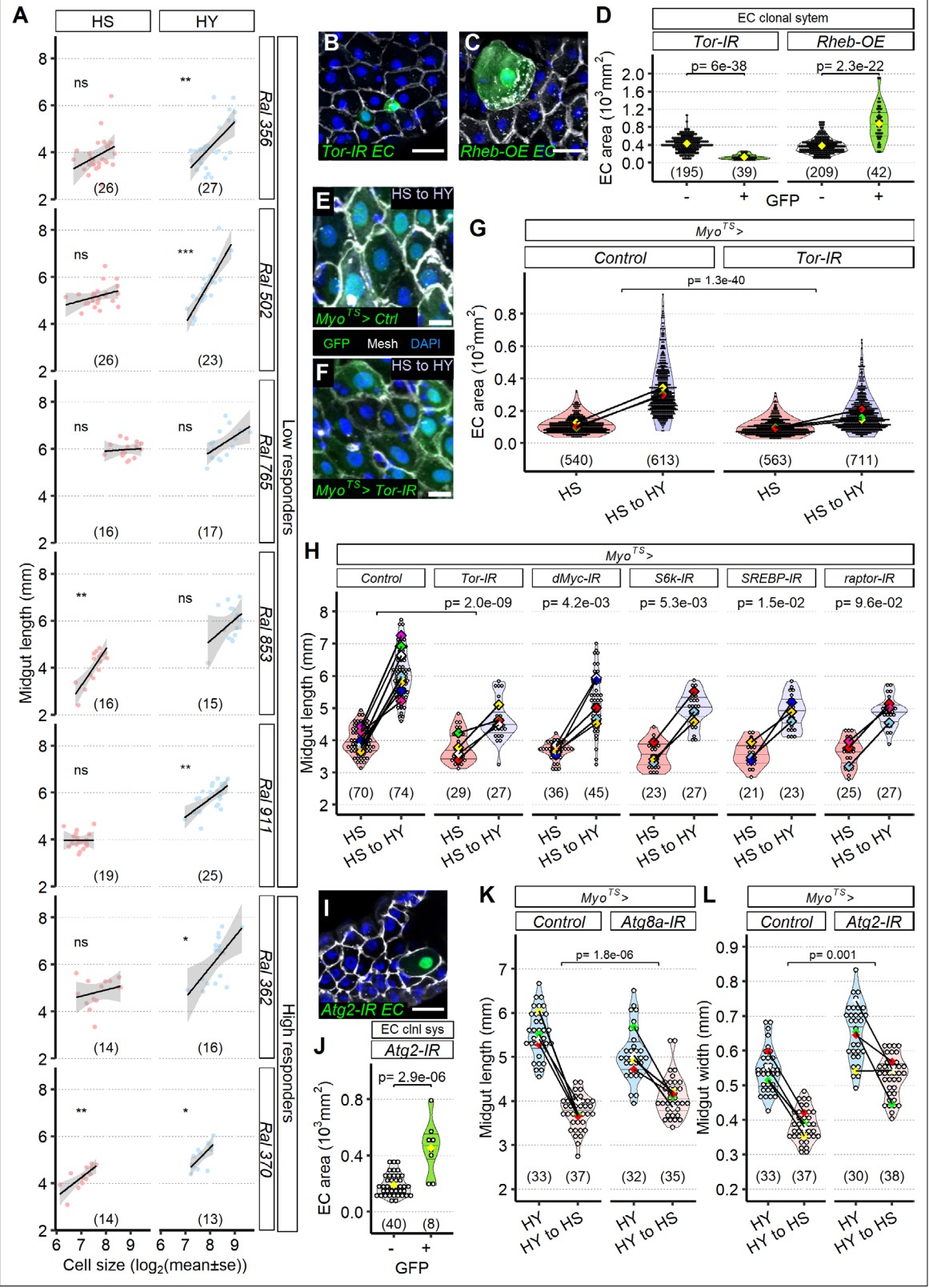

**Figure 7.** Enterocyte (EC) resizing is required for midgut plasticity. (**A**) EC size mostly correlates with midgut length. Quantification of EC area across a selected panel of *Drosophila* Genetic Reference Panel (DGRP) lines comprising high and low responder shows that midgut length mostly correlates with EC cell area, especially in high-yeast (HY) diet. Lines on plot show smoothed splines. Flies were moved on the experimental diets at eclosion and dissected 5 days post eclosion. (**B–H**) The TOR pathway regulates cell size. Representative pictures of single-cell clones (*hsFlp; Act >STOP >Gal4,*

*Figure 7 continued on next page*

*Figure 7 continued*

*UAS-GFP*) suggest that compared to GFP⁻ cells TOR downregulation (*UAS-Tor-IR,* GFP⁺, **B**) results in smaller cells, while TOR hyperactivity (*UAS-Rheb-OE*, **C**) increases cell size. Quantification of clone size in (**D**). Single-cell clones are marked with GFP (green). Dissection for clones was performed 14 days post eclosion. Knockdown of TOR with *Myo^TS*, an EC-specific driver, leads to the increased number of small ECs (**F**) compared to control (**E**); EC-specific GFP is indeed visible in small cells. Quantification of EC area (**G**). Accordingly, blocking TOR pathway components in ECs (*Myo^TS*) inhibits diet-induced midgut growth. Control showed in chart is representative of multiple experiments. Statistical analyses were performed only on appropriate repeat/experiment and comparing interaction between diet and fly line. For (**E–H**), flies were dissected for the first timepoint (HS) at 12 days post eclosion, and then shifted on HY for additional 7 days. (**I, J**) Representative picture utilizing single-cell clonal system suggests that blocking *Atg2* (*hsFlp; Act>STOP >Gal4, UAS-GFP>UAS-Atg2-IR, GFP⁺ cells*) results in bigger ECs compared to control GFP⁻ cells, quantified in (**J**). Dissection for clones was performed 14 days post eclosion. (**K, L**) Blocking autophagy reduces midgut resizing upon shrinkage (HY to HS for 7 days). Blocking *Atg8a* expression with RNAi in ECs (*Myo^TS>UAS-Atg8a-IR*) results in less length shrinkage compared to control midguts. Blocking *Atg2* expression with RNAi in ECs (*Myo^TS>UAS-Atg2-IR*) results in less width shrinkage compared to control midguts. For (**J, K**), flies were dissected at the first timepoint (HY) at 12 days post eclosion, and then shifted on HS for additional 7 days. For the violin/dot plots and boxplots showed in this figure, white dots represent single midgut measurements. Lozenges represent mean of repeats. Violin plots and box plots are color coded according to diets for (**G, H, K, L**) (HS = red, HY = light blue, HY to HS = pink, HS to HY = purple) or to condition for (**D, J**) (GFP⁺ in green, GFP⁻ in gray). Black lines connecting means visualize the interaction between diets. Numbers in parentheses at the bottom of charts indicate sample size. Additional information on the statistics can be found in *Supplementary file 2*. Scale bars are 25 µm for (**B, C, I**) and 10 µm for (**E, F**).

The online version of this article includes the following figure supplement(s) for figure 7:

**Figure supplement 1.** Diet-dependent changes in enterocyte (EC) size correlate with changes in midgut size.

**Figure supplement 2.** The TOR pathway regulates cell size.

**Figure supplement 3.** A model of diet-dependent midgut plasticity.

**Source data 1.** Numeric data for *Figure 7D*.

**Source data 2.** Numeric data for *Figure 7G*.

**Source data 3.** Numeric data for *Figure 7H*.

**Source data 4.** Numeric data for *Figure 7J*.

**Source data 5.** Numeric data for *Figure 7K*.

**Source data 6.** Numeric data for *Figure 7L*.

**Figure supplement 1—source data 1.** Numeric data for *Figure 7—figure supplement 1A*.

**Figure supplement 2—source data 1.** Numeric data for *Figure 7—figure supplement 2C*.

**Figure supplement 2—source data 2.** Numeric data for *Figure 7—figure supplement 2D*.

**Figure supplement 2—source data 3.** Numeric data for *Figure 7—figure supplement 2E*.

an important regulator of diet-dependent EC resizing, which itself is required for midgut resizing following a change in diet.

## Discussion

The intestinal epithelium is the vanguard of an organism's interaction with the external environment, which includes food, microbes, and pathogens (*Miguel-Aliaga et al., 2018*). Consequently, the midgut and its homeostasis are vital for host health and have been shown to determine animal lifespan (*Biteau et al., 2010*). Here, we demonstrate that the midgut is a plastic interface between the host and nutrients, which can dynamically resize throughout the healthy phase of early life in response to changes in the nutritional composition of the diet. To date, the dogma has been that ISC proliferation is the primary driver of midgut responses to environmental variation, but correspondence to resultant midgut size has been neglected. Our data paints a more nuanced picture, showing coincident plasticity in cell gain (ISC proliferation), cell loss, and cell resizing (*Figure 7—figure supplement 2A*). ISC proliferation was surprisingly dispensable for plasticity on the timescale of our experiments, whilst EC size emerges as an essential determinant of nutrition-dependent midgut resizing. Diet-dependent intestinal resizing also occurs in mammals and reptiles, and it will be interesting in future work to characterize whether equivalent mechanisms are also conserved (*Mao et al., 2013*; *Navarrete et al., 2015*; *Petit et al., 2007*; *Thulesen et al., 1999*; *Xie et al., 2020*).

Diet-dependent midgut resizing differentially affected specific midgut regions, producing an overall allometric response to diet. This suggests that either size control in each region depends on different gene networks, as it is the case for thorax and abdomen plasticity (*Lafuente et al., 2018*),

or that the sensitivity of those networks is differentially primed in specific regions. Strikingly, nutrition-dependent midgut resizing varied extensively across lines in the DGRP panel, demonstrating inter-individual genetic variation in the ability of the midgut to respond to diet. Altogether, we propose a model in which diet shapes organismal health not only through the nutrients it provides, but also through changes in the structure of the gastrointestinal (GI) tract that secondarily influence host physiology. Future work will allow us to determine the contribution of the plasticity of the GI tract in physiological responses to nutrition beyond the midgut.

## Multiple nutrients affect midgut resizing

Work using *Drosophila* as a model to dissect the impact of nutrition on host physiology and lifespan demonstrated that proteins and amino acids have a central role in determining animal health. For example, single amino acids such as methionine mediate most of the tradeoff between reproduction and aging (*Grandison et al., 2009*). Selected amino acids also influence diverse aspects of host physiology such as survival of infection or the developmental rate of larvae (*Bing et al., 2018*; *Zhang and Rubin, 2013*). Previous studies focused on elucidating the impact of diet on midgut growth were conducted using either a starved vs. fed paradigm, or a basic diet vs. basic diet with added yeast paste, making it impossible to evaluate the contribution of different nutrients or to disentangle nutrient composition from caloric content (*Choi et al., 2011*; *O'Brien et al., 2011*). In our study, we demonstrate that qualitative variation in isocaloric diets drives midgut resizing, and we make use of nutritional geometry to demonstrate that both diet density and diet composition influence midgut size. While yeast ingestion drives midgut growth, ingested sugar results in midgut shrinkage, demonstrating again that calories per se do not limit midgut growth. Using rescue-like experiments, we investigated the nutrients in yeast that underlie midgut resizing. As expected, protein was required for diet-dependent midgut growth. However, protein alone was not sufficient to induce midgut growth: all components of yeast tested together, including lipids, vitamins, and minerals, were required for growth. These results are somewhat surprising and could be because midgut physiology is different from the physiology of other organs that respond to proteins. When considering ovaries, nutrients such as lipids have accumulated in the fat body during development (*Kühnlein, 2011*) and are thus readily available while proteins are limiting for the generation of new eggs. It is possible that in normal conditions midgut metabolism depends entirely on food for its maintenance while other organs use both stored reserves and nutrients from food. In such a scenario, all nutrients are effectively essential nutrients, acting as both signals and substrates for cell maintenance and production. Our study also demonstrated that changes in texture, as also shown in *Li et al., 2018*, affected midgut size, suggesting that multiple processes and probably a complex gene regulatory network influence midgut plasticity.

## Adaptation to nutrients vs. pathophysiology of sugars

While all the nutrients tested influence midgut size, not all calories are made equal. Specifically, multiple dietary sugars in high amount resulted in midgut resizing. By contrast, an isocaloric diet with higher amount of yeast caused net midgut growth. Two hypotheses could explain the opposite effects of yeast and sugar. If the midgut size is optimized to nutrient availability (*O'Brien et al., 2011*), we would expect that a large midgut is required to potentiate digestion and absorb the complex panel of nutrients sourced from yeast, but not for the simpler task of sugar digestion and uptake. Intrinsic to this model is an expectation that the metabolic cost of growing the midgut is outweighed by diet's nutritional reward, likely the reproductive benefit of yeast consumption (*Skorupa et al., 2008*).

However, an alternative hypothesis is also possible, namely that toxicity of high levels of sugar stunt midgut growth. This hypothesis is supported by our observations that the HS diet promoted a transcriptomic signature of stress, inhibited global translation, and decreased lifespan. This decreased lifespan was not mediated by an increase in fecundity because fecundity is lower in the absence of yeast, suggesting that sugar indeed is deleterious for health. This toxic effect of sugar on midgut tissue renewal could accelerate age-related pathologies such as barrier dysfunction, leading to early death. Of note, while age-associated microbial dysbiosis in the midgut is a key factor that determines fly lifespan, the microbiota was not required for diet-dependent midgut plasticity, suggesting that diet and microbes may additively regulate lifespan. The nature of the stress associated with HS diet remains elusive as most stress-responsive pathways ended up upregulated in midgut of flies fed on the HS diet.

## Is midgut homeostasis only an apparent property?

Several studies have demonstrated that stress or damage inflicted to the midgut result in a higher expression of niche-derived pro-mitotic signals. For instance, microbial infection, ingestion of damaging chemicals, or genetic elimination of ECs all result in a compensatory increase in ISC proliferation (*Amcheslavsky et al., 2009*; *Buchon et al., 2010*; *Buchon et al., 2009b*; *Buchon et al., 2009a*; *Jiang and Edgar, 2009*; *O'Brien et al., 2011*). In addition, maturation of pro-mitotic ligands in ECs is regulated by caspase activity in dying cells (*Liang et al., 2017*), resulting in a direct coupling between cell loss and cell gain. These results together have yielded a model in which the midgut is a homeostatic tissue, keen to maintain a set size that the midgut would return to after challenge. Most importantly, this model suggests that ISC proliferation is coupled to EC loss in a fixed manner. One strong limitation to these models is that they rely mostly on measures of proliferation or local quantifications (e.g., clones) and lack quantification of cell loss. A perfectly homeostatic regulation of the midgut epithelium would imply that cell loss equals cell gain, which leads to a replacement ratio of 1. Using pulse chase experiments to track both the number of cells gained and lost in each midgut on both the HS and HY diet, we found that whether on HS, on HY, or during diet shifts, we never detected a cell replacement ratio approaching 1. Even in flies feeding constantly on the HY diet, more cells were generated in the midgut than lost, resulting in the midgut constantly growing up to 28 days post eclosion. We therefore propose a model (*Figure 7—figure supplement 3A*) in which there is not a constant coupling between cell loss and gain, but instead a flexible connection between ISC proliferation and cell loss as a function of diet, contributing to midgut size. In such a model, homeostasis could be only apparent and emerge from specific conditions where cell gain and loss are equal, which were by chance used in previous studies, rather than inherently programmed. Alternatively, the response to damaging stimuli, such as infection or chemical damage, could trigger a different type of response, less sensitive to variations in ISC niche coupling. It is interesting to note that most experiments documenting tissue repair have been performed on a rich diet, akin to our HY diet, that we find is permissive to growth. It is therefore difficult to tease apart the relative contribution of diet-dependent midgut growth and damage-dependent repair signals in the described homeostatic regrowth. To further integrate these different models, it will be interesting to revisit classical experiments that have led to the notion of epithelial homeostasis of the midgut on multiple diets.

We also observe that while the cell replacement ratio will determine the increase or decrease in midgut cell numbers, and midgut size, similar ratios can occur at very different rates of cell gain and loss. Our results demonstrate that, while diet determines the overall replacement ratio, the 'history' of the diet on which the organisms fed a few days prior also matters. For instance, while we detected a replacement ratio around 0.5 on both midguts staying on HS or transitioning from HY to HS, the rates of gain and loss were higher on the transition. We hypothesize this could originate in the fact that the initial size of the organ differs between experiments (short midgut at the start of HS to HS, long midgut at the start of HY to HS). Future experiments should tackle how past physiological conditions influence later epithelial dynamics. We argue that to understand and capture cell dynamics within a tissue both the replacement ratio and the rates of cell gain and cell loss need to be measured. In the current study, estimating only ISC proliferation would have led to an incomplete picture of tissue turnover.

Finally, these results were obtained by keeping flies constrained to an imposed specific diet, to tease apart the impact of different nutritional components on the midgut. However, a caveat of this conclusion on midgut homeostasis is that, in the wild, flies do not have a fixed diet and may vary their feeding among different available nutrients, possibly alternating periods of midgut growth with periods of midgut shrinkage. We do not know how this process would affect midgut homeostasis, but it may (or may not) result in an overall homeostatic replacement rate.

## Contributions of EC size and cell numbers to midgut resizing

Most studies of midgut size have focused on stem cell proliferation, leading to the conclusion that an increase in midgut size is concomitant to an increase in ISC activity and symmetrical division. However, it remains unclear whether the increase in mitosis is causal or even required for resizing. During development, imaginal discs grow through a combination of proliferation and cell size growth, with growth being dominant over, and compensating for, cell division defects (*Neufeld et al., 1998*). We found that diet-dependent midgut growth is concomitant to both increased cell number (connected to increased

ISC proliferation) and increased EC size (*Figure 7—figure supplement 3A*). In addition, blocking ISC proliferation did not prevent midgut growth on the HY diet, suggesting that ISC proliferation can be dispensable for midgut resizing. In such midguts, the increase in EC size was higher than in control intestines, compensating for decreased cell number. As both proliferation and size increase on the HY diet, and as EC size dominates, this raises the question of what the main driver of organ size is. One answer came from looking at lines from the DGRP panel. In these 11 wild-caught isolines, almost no positive correlation was observed between proliferation rate and midgut size on either diet (only one line made exception and only on HY diet). In contrast, a larger number of lines showed a correlation between midgut length and EC size (showing an overall strong correlation between increase in EC size and increase in midgut length when comparing HY/HS), indicating that EC size may drive midgut resizing. It remains possible that this principle of cell size would vary amongst lines, especially in lines that resize strongly in response to diet as there is a physical limitation to the size a cell can reach. It is also possible that the apparent dominance of size over proliferation is just a consequence of the size difference between progenitors and mature cells. It would require a very high number of mitoses to accumulate enough small progenitors to compensate for one EC. Accordingly, forced ISC over proliferation, for example, by *Ras$^{V12}$* overexpression (*Buchon et al., 2010*), does not increase midgut size, but rather shortens the midgut due to epithelial multilayering. This suggests that proliferation and cell size need to be properly coordinated to lead to normal changes in organ size. This hypothesis agrees with a recent study demonstrating a balance between stem cell mitosis and EC nucleus growth in the midgut of *Drosophila* (*Tamamouna et al., 2020*). Additionally, we cannot discount a role for EBs in organ regulation. Regulation of EBs could be, for example, a critical step in setting up the size for new ECs, and EBs are known to be able to differentiate at different rates, or undergoing apoptosis (*Reiff et al., 2019*). We therefore propose that the organ reaches a set size based on diet, suggesting that the notion of a 'size meter,' introduced in developmental biology, could also apply to adult tissues (*Shingleton, 2010*). However, the nature of such a 'size meter' remains elusive, and its characterization will be complicated by the fact that it is nutrient dependent. Moreover, in physiological conditions midgut size may just be a function of midgut digestive capability. In this context, it is possible that while still being able to grow, a midgut expanding only through EC size is not as efficient as a normally growing midgut, or more prone to deregulation in later phases of life.

## Translation control as a general regulator of ISC coupling to its niche

Our results suggest that high dietary sugar levels lead to a decrease in coupling between ISC proliferation and the expression of pro-mitotic signals (*Figure 7—figure supplement 3A*). ISCs seem to be still capable of proliferating on the HS diet, as expression of *Ras$^{v12}$* and *Tor-DER* led to a strong increase in proliferation, comparable to midguts on the HY diet. Uncoupling is mediated by a *Gcn2*-dependent stress response that decreases global translation in the midgut, leading to a decrease in production of niche signals. A caveat of this conclusion is that we relied on pH3 as the sole readout for these experiments. It is possible to speculate that, despite having increased pH3 counts, ISCs may be cycling slower or entertaining different modalities of division (symmetric vs. asymmetric), which would result in a low proliferative ability despite the increased pH3 count. Interestingly, infection with lethal dose of *Pseudomonas entomophila* leads to similar translation blockage, and rescuing translation alleviates pathogenicity (*Chakrabarti et al., 2012*). These together demonstrate that global translation levels can affect ISC coupling to its niche in multiple physiological contexts. We therefore propose a general model of ISC control where stress and nutrients alter both transcription and translation of niche-derived signals, ultimately leading to ISC proliferation.

## Conclusions

In this study, we find that the midgut plastically resizes in response to shifts in diet composition. This resizing is mediated by the balance of cell gain and loss, stem cell-niche coupling, and EC size. Dietary manipulation also challenges our view of epithelial dynamics in a physiological context, demonstrating that nutrients affect the rates of both cell loss and cell gain independently. Importantly, diet-dependent midgut plasticity is a variable trait in the population, opening the possibility that interindividual differences in nutritional physiology originate in variations in midgut plasticity. Our work points to intestinal plasticity and its regulation by diet as an important, overlooked, and complex phenomenon likely to impact health and disease.

## Materials and methods

### Key resources table

See *Appendix 1—key resources table*.

### Fly stocks

*Drosophila* stocks were maintained at room temperature (~23 °C) on yeast-cornmeal medium (pre-experiment diet, *Supplementary file 1*) or at 18 °C in a 12:12 hr light/dark cycle incubator. Canton-S (Cs) (BDSC: 64349), a wild-type inbred line, was used as a wild-type for all experiments not involving transgenic constructs or the DGRP panel. *Gal4* drivers used were 'w⁻;Esg-Gal4; UAS-GFP, tub-Gal80$^{TS}$' (*Esg$^{TS}$*, progenitor-specific, *Micchelli and Perrimon, 2006*); 'Esg-Gal4, tub-Gal80$^{TS}$, UAS-mcherry-CD8" *Esg$^{TS}$*, progenitor-specific, *Nagy et al., 2018*; 'Esg-Gal4, UAS-GFP, tub-Gal80$^{TS}$; Act>STOP >Gal4,UAS-flp' (*Esg$^{F/O}$*, progenitors + marked lineage, *Jiang et al., 2011*); 'w⁻; Myo1A-Gal4, UAS-GFP, tub-Gal80$^{TS}$; upd3-lacZ' (*Myo$^{TS}$*, EC-specific, *Buchon et al., 2010*); 'Actin5C-Gal4/Cyo; tub-Gal80$^{TS}$,UAS-GFP' (*Act$^{TS}$*, whole fly); '5966GS' (EC-EB-specific, gene-switch RU486-dependent, *Guo et al., 2013*); 'hsFlp; Act>STOP >Gal4,UAS-GFP' (single-cell clonal system, *Ito et al., 1997*); 'P{ry[+ t7.2] = hsFLP}12,y[1] w[*]; P{w[+ mC] = UAS GFP.S65T}Myo31DF[T2]; P{w[+ mC] = Act5 C(-FRT)GAL4. Switch.PR}3/TM6B, Tb[1]' (*ActGS*, this is P{Act5C(FRT.y[+])GAL4.Switch.PR}3 with the y[+] FRT cassette removed by FLP recombination, BDSC 9431). UAS lines used were (with reference or Bloomington *Drosophila* Stock Center code in parenthesis) UAS-Histone2B-RFP (*Mayer et al., 2005*), UAS-Gcn2-IR (67215), UAS-LK6-IR (60003), UAS-EGFR-IR (60012), UAS-Tor-IR (34639), UAS-Myc-IR (36123), UAS-raptor-IR (34814), UAS-S6k-IR (41702), UAS-SREBP-IR (34073), UAS-RagA-B-IR (34590), UAS-RagC-D-IR (32342), UAS-Atg8a-IR (34340), UAS-Atg2-IR (34719), UAS-Ras85D$^{V12}$ (64195), UAS-Tor-DER ('yw; Pw+; (UAS torD-DER)' II RJH430, gift from M. Freeman), UAS-upd3-OE (gift of M. Crozatier) (*Brown et al., 2001*), UAS-spi-SEC (58436), UAS-Rheb-OE (9690), UAS-rpr-OE (5823), UAS-InR-DN (8252, the line was backcrossed six times into an outbred population with a w⁻$^{Dah}$ background). Reporter lines used: GBE-Su(H)-lacZ (*Micchelli and Perrimon, 2006*), thor-lacZ (9558). DGRP: we used 188 lines; measurements can be found in *Figure 1—source data 2* (*Mackay et al., 2012*).

### Experimental design

Mated female flies were used for all experiments, except for *Figure 1—figure supplement 1C*. Flies, of which the progeny would be used for experiments, were allocated in equal numbers (~20 females and ~10 males) in tubes containing the pre-experiment diet and allowed to seed the tube for 2 days in a 12:12 hr light cycle at 25° C incubator. F1 progeny were collected every 3 hr on the day of eclosion and immediately transferred to the experiment diets to avoid confounding effects of adult feeding on the pre-experiment diet. To generate conditional knockdowns/overexpressions with the *Gal4–Galt80$^{TS}$* system (TARGET; *McGuire et al., 2003*), crosses were made using ~15 female flies and 5 males and transferred during development to a 12:12 hr light cycle and 18 °C incubator on the pre-experiment diet. *Gal4–Galt80$^{TS}$* parents were always females and were crossed to male flies carrying UAS constructs. Parents were removed after 5 days to control for fly density of the F1 progeny. The F1 progeny were collected every 3 hr on the day of eclosion and transferred to experimental diets, where they were kept at 18 °C for 5 days to allow for proper midgut development. F1 progeny were then moved for 7 days at 29 °C to allow for transgene induction to take effect. This timepoint was considered as day 0; on this day, some flies were dissected from each group and the remaining were shifted from HS to HY, or from HY to HS, and kept on the new diet for 7 days until dissection. *Gal4–Galt80$^{TS}$* flies crossed to *Cs* were used as experimental control, except for experiments involving *UAS-InR-DN*, where w⁻$^{Dah}$ was crossed to *Gal4–Galt80$^{TS}$* flies and used as a control. For experiments involving over-expression of *UAS-Ras$^{V12}$* and *UAS-Tor-DER*, F1 flies were kept during development at 18 °C to keep the TARGET system off. At eclosion, flies were transferred on HS diet for 2 days and split afterward on HS and HY for 3 days. Flies were then shifted at 29 °C to induce transgenes, and midgut were dissected after ~12–16 hr induction. For experiments involving overexpression of *UAS-upd3-OE* and *UAS-spi-SEC*, we followed the same protocol as for *UAS-Ras$^{V12}$*, but flies were dissected after 3 days at 29 °C. For experiments involving genetic manipulation with *5966GS*, we utilized the same timing as the experiments performed with the *Gal4–Galt80$^{TS}$* system but supplying food with RU486 to induce UAS-driven transgenes, and flies were kept at room temperature for the whole duration of the experiments. 100 μL of a 5 mg/mL solution of RU486 (Cat# M8046, Sigma-Aldrich) in 80% ethanol was

added on top of the food and dried for at least 16 hr to allow for vehicle evaporation, as previously published (*Biteau et al., 2010*). Flies of identical genotype were used as control for this experiment but were moved instead onto vials containing only the vehicle solution (80% ethanol), which were manipulated in the same way as the vials containing RU486.

## Food production

Food was cooked in an Erlenmeyer flask (Corning, Glendale, AZ) on a hot plate with a magnetic stirrer (VWR, Radnor, PA). Half of the total volume of water together with agar (SKU# 41054, Mooragar, Rocklin, CA) was brought to boiling temperature while stirring. The water-agar solution was then removed from the heat and cooled while stirring. Yeast (Cat# 903312, MP Biomedicals, Irvine, CA), sugar (Walmart, Bentonville, AR), and other diet-specific ingredients were added at this point unless temperature labile. More water, up to 90% of the final volume, was then added. Food was allowed to cool to ~60 °C, before adding acid mix (in the amount specified in *Supplementary file 1*), and water to reach the final volume. Acid mix recipe: for 1 L, 418 mL of propionic acid (CAS # 79-09-4, EMD Millipore, Darmstadt, Germany), 41.5 mL of phosphoric acid (Cat# 2796-16, Macron, Avantor, Center Valley, PA), 540.5 mL of water. Food was cooled to ~40 °C before being aliquoted (8 mL per vial [Cat# 75813-162], VWR). Other ingredients used: inulin (Cat# CAAAA18425-09, VWR), cellulose (Cat# IC19149980, VWR), pectin (Cat# P9135, Sigma-Aldrich), lard (land o lakes, Walmart), casein (Cat# C5679, Sigma-Aldrich), AA mix (TD.10473 and TD.110036, Harlan Laboratories, Inc, IN; *Lee and Micchelli, 2013*), vitamin and mineral mix (TD.10475, Harlan Laboratories; *Lee and Micchelli, 2013*), cholesterol (Cat# C8667, Sigma-Aldrich), moldex (Cat# QB-A611-0572-159, Neta Scientific, Hainesport, NJ), yellow cornmeal (Aunt Jemima, Walmart), glucose (Cat# A16828, Alfa Aesar, Tewksbury, MA), maltose (Cat# M5885, Sigma-Aldrich), fructose (Cat# F0127, Sigma-Aldrich), arabinose (Cat# 80502-266, VWR), and sorbitol (Cat# 76177-308, VWR). Bloomington standard diets were made with nutri-fly mixes: BL cornmeal (Cat# 66-112, Genesee Scientific, El Cajon, CA) and BL Molasses (Cat# 66-116, Genesee Scientific). Refer to *Supplementary file 1* for a detailed list of ingredients for each diet.

## Immunohistochemistry and histology

After dissection, *Drosophila* midguts were fixed in 4% paraformaldehyde (Electron Microscopy Sciences, Hatfield, PA, Cat# 15713S ) in 1× PBS (Cat# 003002, Thermo Scientific, Waltham, MA) for 45–90 min and subsequentially washed three times with 0.1% Triton X-100 (Cat# T8787, Sigma, St. Louis, MO) in PBS. Midguts to be immunostained were then incubated for an hour in blocking solution (1% bovine serum albumin [Cat# 12659, EMD Chemicals, San Diego, CA], 1% normal donkey serum [RRID:AB_2337258, Jackson Laboratories, West Grove, PA] in PBS). Overnight primary antibody staining was performed at room temperature in the blocking solution. Midguts were washed three times with 0.1% Triton X-100 in PBS, and overnight secondary antibody staining was performed in blocking solution. Primary antibodies used: rabbit anti-pH3 (1:1000, Cat# 06-570, EMD Millipore-Sigma), mouse anti-pH3 (1:1000, Cat# 05-806, EMD Millipore-Sigma), rabbit anti-β-galactosidase (1:1000, Cat# A11132, Invitrogen, Carlsbad, CA), mouse anti-Prospero (1:100, Cat# MR1A, DSHB), rabbit anti-mesh (1:2000, Cat# 995-1, gift from Mikio Furuse, *Izumi et al., 2012*), rabbit anti-p-eIF2α (1:500, Cat# 3398, Cell Signaling Technologies, Danvers, MA), and mouse anti-puromycin (1:100, Cat# PMY-2A4, DHSB). Secondary antibodies used: donkey anti-rabbit Alexa 555 (1:2000, Cat# A31572, Thermo Fisher), donkey anti-mouse Alexa 555 (1:2000, Cat# A31570, Thermo Fisher), donkey anti-rabbit Alexa 647 (1:2000, Cat# A31573, Thermo Fisher), and donkey anti-mouse Alexa 647 (1:2000, Cat# A31571, Thermo Fisher). DNA was stained with DAPI in PBS and 0.1% TritonX (1:50,000 from a stock with 10 mg/mL concentration, Cat# D9564, Sigma-Aldrich) for 30 min, and samples received a final three washes in PBS before mounting in Citifluor AF1 antifade medium (Cat #17970-100, Electron Microscopy Sciences). Imaging was performed on a Zeiss LSM 700 fluorescent/confocal inverted microscope (Zeiss, Oberkochen, Baden-Württemberg, Germany).

## Midgut length and width measurements

To measure midgut length, width, and area, tiled images of entire midguts were acquired with fluorescence imaging with a 10× objective and assembled into single images of each midgut with Zen imaging software (Zeiss). Midgut length, width, and area were measured with FIJI (*Schindelin et al.,*

*2012*) by drawing of a spline line along the midgut (length), or perpendicularly to it (width) (see *Figure 1D* for an example), and drawing a polygonal selection along the region of interest (area).

## Nano-CT scan

Flies were treated as previously published (*Mattei et al., 2015*) for image acquisition. Images were visualized with OsiriX DICOM viewer (*Rosset et al., 2004*). To create 3D reconstructions, organs were visually identified and manually marked for each image composing the nano-CT scan stack. Video clips were rendered with OsiriX and assembled using Adobe Premiere Pro.

## Feeding assay

We used a modified version of the method described in *Min and Tatar, 2006* to allow survey of solid food passage. FD&C1 blue dye (Cat# 700010-048, VWR), which is a dye not affected by pH and digestive enzymes (*Shimada et al., 1987*; *Tanimura et al., 1982*), was added to solid food. The volume of food ingested is proportional to the concentration of dye found in the feces. Pools of 20 female flies and 5 males were placed for 24 hr in a 50 mL tube, the cap of which contained 2 mL of the diet of interest, dyed with 0.25% of FD&C1 blue dye. Feces deposited on the side of the falcon tube were resuspended with 2 mL of water and the optical density of the resultant solution was measured at 625 nm with a SmartSpec 3000 Spectrophotometer (Bio-Rad, Hercules, CA). The same flies were subjected to multiple (five) consecutive measurements for each repeat and showed similar measurements at each timepoint. These consecutive measurements also ensured that midgut capacity did not interfere with the amount of measured egested food since the midgut will be already filled with blue food.

## Generation of germ-free flies

Eggs were suspended in 1× PBS and successively rinsed in 70% EtOH for 1 min and dechorionated using 10% bleach for ~10 min. Eggs were then transferred under a sterile laminar flow hood, where they were rinsed three times with sterile ddH$_2$O. The eggs were finally transferred into sterile vials with sterilized fly food. Flies were tested for the presence of bacteria after each experiment by plating homogenates on MRS agar plates.

## DGRP analysis

Response indices were calculated as per-line difference in mean length, divided by whole-DGRP difference in mean length. Broad sense heritability of the response was calculated as the proportion of variance of the ratio of midgut length from each diet within line (as given by the estimate of the random effect 'DGRP ID') over the total variance in ratio (given by variance within line summed with residual variance). To determine how many DGRP lines responded to the diet treatment, we tested for the difference in total midgut length between the two diets for each of the DGRP line with a t-test. The p-value of each of the comparisons was corrected for multiple tests using a false discovery rate method.

## Genome-Wide Association Study (GWAS) analysis

The genetic diversity of the DGRP lines comprises about 4 million SNPs. We selected the SNPs (n = 2,174,256) for our association study based on two criteria: (1) avoid a complete collinearity (possibly confounding) between alleles and Wolbachia status (i.e., we excluded cases where one allele corresponds to Wolbachia infection and the other to an uninfected status); and (2) we had enough lines per treatment to run the model. Prior to each test, we therefore calculated a two-by-two matrix with Wolbachia status and allele identity (i.e., W$^+$/allele1, W$^-$/allele1, W$^+$/allele2, W$^-$/allele2) summarizing the sum of lines for each category. We further included in our association only the SNPs where at least three of the categories had five lines. All the analyses were performed in R. To test for the response to diet, we next estimated the significance of the difference between alleles of the difference in midgut length on each diet at each selected SNP. We used a Generalized Linear Mixed Model (GLMM, function HLfit from the R package 'spaMM'). The model was as follows: *'DifferenceTotalLength'~ DifferenceGenotypeWeight + SNP + (1|wolbachia/DGRP_lines) + (1|block)*. The variable 'Block' accounted for group of flies dissected in the same day (it would not be possible to dissect the whole DGRP for one repeat in 1 day), and the identity of the lines was accounted for as random effect following a

Gaussian distribution. We compared the log likelihood of the complete model to a model lacking the main effect SNP to calculate p-values. We performed a likelihood ratio test in R as follows: pchis-q(Chi2_LRT_snp, df = 1, lower.tail = F) where *Chi2_LRT_snp* is 2 × (log likelihood complete model – log likelihood reduced model).

Candidate SNPs had p-values between $10^{-6}$ and $10^{-9}$. To understand the difference between alleles at a given candidate SNP, we characterized the implication of the mutation on gene function (e.g., missense mutation, point mutation in regulatory sequence, etc.). The DGRP SNP positions are provided for the version 5 of the *D. melanogaster* genome. We then converted the positions in the equivalent for version 6 with the convert tool from Flybase. The characterization of the mutation at each candidate SNP was then provided using the Variant Effect Predictor (VEP) from the website Ensembl (http://www.ensembl.org/info/docs/tools/vep/index.html). Full results are provided (*Figure 1—figure supplement 3A*, *Figure 1—source data 2* and *Figure 1—source data 2*), and highlighted candidates were selected based on the shape of the peak in the Manhattan plot and the function provided by VEP. In order of priority, our highlighted candidates had either non-synonymous mutations, mutations in the 5′ and 3′ UTR regions, or mutations in introns (which could be located in gene enhancers).

Cell counts pH3$^+$ cells were counted directly through the fluorescent microscope eyepiece using a 20× objective. To score numbers of each cell type, *Esg$^{TS}$* females were crossed to *GBE-Su(H)-lacZ* males. Progeny carrying both constructs were dissected and cells were labeled with anti-Prospero (EE), anti-βGAL antibodies (EB), DAPI (all cells, polyploid cells counted as EC), and the Esg-specific GFP expression (progenitor cells). Stacked images of region 4 of the midgut (*Buchon et al., 2013*) encompassing one hemisphere of the midgut along the dorsal/ventral axis were acquired with confocal microscopy with a 20× objective using Zen imaging software (Zeiss). Stacked images were subjected to orthogonal projection, resulting in one image comprising all cells in half of a midgut along the dorsal/ventral axis (Zen/FIJI). The number of Esg$^+$ and GBE-Su(H)$^+$ cells was manually counted. The number of DAPI$^+$ cells with EC comparable size and Prospero$^+$ cells was counted through a semi-automated macro that we developed in FIJI. ISC number was determined by subtracting GBE-Su(H)$^+$ cells from the total number of Esg$^+$ cells. EB number was determined by scoring GBE-Su(H)$^+$ cells. EC number was determined by counting the DAPI stained nuclei within the proper size range. EE cells were determined by scoring Prospero$^+$ cells. The area of the midgut in the acquired image was also measured to determine cell density. Cell density was then multiplied by total area of the region to obtain the total number of cells within the said region. This number was multiplied by 2 to account for only half of the midgut having been imaged. Total area of the region was obtained as described in the section 'Midgut length and width measurements'.

## Cell size measurement

For EC area measurements, midguts were stained with anti-mesh antibody to highlight cell contours. Z-stacks were acquired with a 20× objective using Zen imaging software (Zeiss). Stacked images were manually measured using the polygon selection tools in FIJI. For each cell, the Z-plane with the larger area was selected for the measurements. To avoid selection bias, the measured cells were adjacent to each other, rather than picked at random throughout the midgut (staining quality permitting). Roughly 30 cells per midgut were measured. For EC height measurements, Z-stacks with a 1 µm step size were acquired from *Myo$^{TS}$>Canton* S, which resulted in EC cytoplasm being labeled with GFP. Z-stack position at the top and bottom of single ECs was measured, and EC height inferred. Roughly 30 ECs per midgut were measured.

## Ploidy measurement

Dissected midguts, with proventriculus cut off (25 for sample), were frozen upon dissection in a –80 °C freezer in Grace's medium (Cat# 11605094, Thermo Fisher Scientific) with added 10% DMSO (Cat# 80058/040, VWR) until dissociation. On dissociation, samples were thawed on ice. Midguts were spun down, and freezing media was removed. 2 mL of ice-cold homogenization buffer (HB) was added with the midguts to a Dounce homogenizer. Guts were subjected to 50 strokes with loose pestle. After 5 min on ice, 10 additional strokes were performed. After five additional minutes, 20 strokes with tight pestle were performed. Samples were then run through a 70 µm cell strainer and spun down for 5 min at 500 g at 4 °C. Supernatant was then removed and 600 µL of Wash buffer were added, and nuclei were resuspended in it. Samples were centrifuged again for 10 min at 750 g at 4 °C

and filtered a second time through a 40 μm cell strained. Wash solution was removed and 300 μL of PBS with DAPI (10 μg/mL) were added to each sample. Samples were run in an Aria Fusion sorter at Cornell Institute of Biotechnology – FACS core. Samples were analyzed with FCS express 6 (De Novo software). Samples were gated to remove doublets and debris, and ploidy peaks percentages were measured. 6 X HB composition: 30 mM $CaCl_2$ (Cat# 122950, Beantown Chemical, Hudson, NH), 18 mM $Mg(Ac)_2$ (Cat# 12225, Alfa Aesar), 60 mM Tris pH 7.8 (Cat# 0497, VWR), 0.1 mM PMSF (Cat# A0999, AppliChem, Council Bluffs, IA), and 1 mM β-mercaptoethanol (Cat# 0482, Amresco). 1 X HB: 6HB to 1HB, 320 mM sucrose (Cat# 902978, MP Biomedicals), 0.1 mM EDTA (Cat# E177, VWR), 0.1% NP40 (Cat# 19628, United States Biochemical, Cleveland, OH), and protease inhibitor tablet (Cat# A32963, Thermo Fisher Scientific). Wash buffer: 10 mM Tris-HCl pH 7.4, 10 mM NaCl (Cat# 470302-512, Ward's Science, St. Catharines, ON, Canada), 3 mM $MgCl_2$ (Ca# 442611, EMD Chemicals), 0.1 % Tween-20 (Cat# P1379, Sigma-Aldrich).

## Cell loss assay

*5966*GS>*Histone2B-RFP* flies were fed either HS or HY diet for 5 days from eclosion. RU486 supplemented food was produced as previously described. Flies were then fed on HS and HY supplemented with RU486 for 3 days, moved back on HS and HY without RU486 for 2 days to allow for its elimination, and dissected (day 0). Flies were subsequently shifted to HS and HY for 14 days, when they were dissected, to create the four conditions analyzed (HS to HS, HS to HY, HY to HY, HY to HS). Confocal Z-stacks of half a midgut hemisphere were acquired with a 20× objective to determine the number of RFP$^+$ and RFP$^-$ cells. The area of the midgut in the acquired image was also measured to determine cell density. Tiled images were acquired with a 10× objective to determine regional midgut area, which was measured with FIJI as previously described (see section 'Midgut length and width measurements'). Density was multiplied by midgut regional area and then by 2 (to account for only half midgut being imaged) to determine the total cell number per midgut region. To calculate turnover, we divided the new cells made by the number of lost cells. To find the total number of lost cells, we had to factor both RFP$^+$ and RFP$^-$ lost cells (not 100% of polyploid cells was marked at day 0). To calculate the RFP$^+$ lost cells, we subtracted ending RFP$^+$ cell number from initial RFP$^+$ cell number (initial RFP$^+$ - ending RFP$^+$). The number of lost cells that were initially unmarked was then calculated by multiplying the amount of initial unmarked cells (initial RFP$^-$) by the ratio of RFP$^+$ leaving [1 – (ending RFP$^+$/initial RFP$^+$)]. This resulted in the formula describing number of unmarked cells lost to be: initial RFP$^-$ * [1 – (ending RFP$^+$/initial RFP$^+$)]. Addition of these two numbers of cells lost (marked and unmarked) gave us the total number of lost cells: (initial RFP$^+$ – ending RFP$^+$) + (initial RFP$^-$ * [1 – (ending RFP$^+$/initial RFP$^+$)]). New cells were calculated by subtracting from the ending number of RFP$^-$ cells, the amount of initial RFP$^-$ cells – the amount of initial RFP$^-$ cells lost (as previously calculated). This results in new cells being calculated with the following formula: ending RFP$^-$ – (initial RFP$^-$ – ((initial RFP$^-$ * [1 – (ending RFP$^+$/initial RFP$^+$)]))). The final resulting formula is as follows: ending RFP$^-$ – (initial RFP$^-$ – ((initial RFP$^-$ * [1 – (ending RFP$^+$/initial RFP$^+$)])))/((initial RFP$^+$ – ending RFP$^+$) + (initial RFP$^-$ *[1 – (ending RFP$^+$/initial RFP$^+$)]).

## RNA-seq generation and analysis

50 midguts for each condition/repeat (three repeats) were dissected at eclosion and on days 1, 2, 3, and 5 post eclosion, immediately transferred to ice-cold Trizol (Cat# 15596018, Life Technologies, Carlsbad, CA), and homogenized. Total RNA was isolated using a hybrid Trizol-Rneasy (Cat# 74106, Qiagen, Hilden, Germany) protocol, as previously published (*Houtz et al., 2019*; *Troha et al., 2018*). RNA was quantified with Qubit (Thermo Fisher) and quality checked via a fragment analyzer at Cornell genomic facility. QuantSeq 30 mRNA-Seq Library Prep Kit FWD (Cat# 015.2X96, Lexogen, Vienna, Austria) was utilized to prepare 3′ end RNA-seq libraries. Libraries were quality checked before pooling and sequencing with the Illumina Nextseq 500 platform by the Biotechnology Resource Center (BRC) Genomics Facility at the Cornell Institute of Biotechnology (http://www.biotech.cornell.edu/brc/genomics-facility). 5–6 million reads were sequenced per sample, which approximately equals a 20× coverage by conventional RNA-seq. Libraries were checked by FastQC, adaptors trimmed by CutAdapt (*Martin, 2011*), aligned to the fly genome using Tophat (*Trapnell et al., 2009*), and counted with HTSeq (*Anders et al., 2015*). Enumerated reads were analyzed in R (3.3.1)/BioConductor using

DESeq2, testing the effect of diet with a likelihood ratio test, using day 3 and 5 data. GO analysis was performed in R with topgo. Data have been submitted to ArrayExpress (E-MTAB-10812).

## Translation assay

General translation was measured with a puromycin incorporation assay (*David et al., 2012*; *Deliu et al., 2017*). Cs flies fed either HS or HY diet for 5 days were moved to HS or HY diet, which had 150 µL of 5 µL/mL of puromycin (Cat# P8833, Sigma-Aldrich) in water added to the surface of the food just prior to the experiment. Flies were dissected after 3 hr, following standard immunochemistry protocol as previously described.

## Single-cell genetic manipulation

*hsFlp; Act>STOP >Gal4,UAS-GFP* flies have a base leakiness in their activity. This system was crossed with UAS constructs of interest to generate GFP marked single cells expressing the UAS construct of interest in an otherwise normal midgut. We acquired images of these midguts stained with anti-mesh antibody to mark cell membranes and measure cells size. For each GFP⁺ clone (driving the transgene of interest), we quantified ~5 GFP⁻ control cells (not driving the transgene of interest) from the same image.

## Statistical analysis

We provide the complete statistical formula and raw data used for each experiment in the R mark-down file (*Supplementary file 2*; https://dduneau.github.io/Bonfini_eLife_2021/Bonfini_eLife_2021.html). In brief, we mostly used generalized linear mixed models (function fitme from the R package 'spaMM'; *Rousset and Ferdy, 2014*). To compare the difference between factors, such as diet or genotype, the model was, for example, as follows*: Gut_length~ Diet + (1|Repeat)*. To compare the difference in response, such as the midgut length on each diet for different genotypes, the model was as follows: *Gut_length~ Diet + Genotype+ Diet:Genotype (1|Repeat), where 'Diet:Genotype' represents the interaction between the variables*. The variable 'Repeat' describes the experimental replication and was accounted for as random effect following a Gaussian distribution. We then tested the difference between main effect (or the difference in response) by comparing the log likelihood of the complete model to a model lacking the main effect (or the interaction) to calculate p-values (displayed in the figures). We performed a likelihood ratio test in R as follows: pchisq(Chi2_LRT, df = 1, lower.tail = F), where *Chi2_LRT* is 2 × (log likelihood complete model – log likelihood reduced model). Normal distribution and homoscedasticity of the residuals were tested with Shapiro–Wilk normality tests and Brush–Pagan tests, respectively. In many cases, the response variable was log transformed to improve model fit. To characterize differences between several conditions, general linear hypotheses tests were applied, using a Tukey post hoc pairwise comparisons (i.e., fitting an adequate model followed by a glht function -alpha = 0.05 from the package multcomp in R; *Hothorn et al., 2008*). Pearson correlation tests were performed (function *cor.test* from the default R stat package) for *Figures 6A and 7A*. The p-values were corrected and adjusted by the false discovery rate correction (*Benjamini and Hochberg, 1995*).

## Acknowledgements

We thank M Furuse for reagents. We thank Rachel L Fay and Yunan Nie for help with project-related experiments not included in the manuscript. We thank the Cornell Institute of Biotechnology imaging (NanoCT), genomic facilities (RNA-seq), and FACS core for help in setting up protocols and general assistance. We thank the Lazzaro lab for access to equipment and reagents.

# Additional information

## Funding

| Funder | Grant reference number | Author |
|---|---|---|
| National Institutes of Health | 1R21AG065733-01 1R01AI148541-01A1 | Alessandro Bonfini Jonathan Revah Xi Liu Philip Houtz Nicolas Buchon |
| National Science Foundation | IOS-1656118 IOS-1653021 | Alessandro Bonfini Jonathan Revah Xi Liu Philip Houtz Nicolas Buchon |
| UK Research and Innovation | MR/S033939/1 | Adam J Dobson |
| Agence Nationale de la Recherche | ANR-10-LABX-41 ANR-11-IDEX-0002-02 | David Duneau |

The funders had no role in study design, data collection and interpretation, or the decision to submit the work for publication.

## Author contributions

Alessandro Bonfini, Conceptualization, Data curation, Formal analysis, Investigation, Methodology, Validation, Visualization, Writing - original draft, Writing - review and editing, conceived the study, designed, and performed most of the experiments, contributed to dissections for DGRP, performed RNA-seq analysis, performed image analysis and organs rendering for NanoCT scan, wrote original manuscript, contributed to dissections for DGRP, contributed to dissections for DGRP, contributed to dissections for DGRP; Adam J Dobson, Conceptualization, Data curation, Formal analysis, Investigation, Methodology, Validation, Visualization, Writing - original draft, Writing - review and editing, conceived the study, contributed to dissections for DGRP, performed GWAS analysis, performed RNA-seq analysis, performed experiments for NanoCT scan and image acquisition, contributed to dissections for DGRP, contributed to dissections for DGRP, contributed to dissections for DGRP; David Duneau, Data curation, Formal analysis, Validation, Writing - review and editing, performed GWAS analysis, performed statistical analyses and R-code improvements; Jonathan Revah, Xi Liu, Philip Houtz, Investigation, Writing - review and editing, contributed to dissections for DGRP, contributed to dissections for DGRP, contributed to dissections for DGRP; Nicolas Buchon, Conceptualization, Data curation, Formal analysis, Funding acquisition, Investigation, Methodology, Project administration, Resources, Supervision, Validation, Visualization, Writing - original draft, Writing - review and editing, conceived the study, performed RNA-seq analysis, contributed to dissections for DGRP, wrote original manuscript, supervised the project and funding., contributed to dissections for DGRP, contributed to dissections for DGRP, contributed to dissections for DGRP

## Author ORCIDs

Alessandro Bonfini (iD) http://orcid.org/0000-0001-6642-8665
Adam J Dobson (iD) http://orcid.org/0000-0003-1541-927X
David Duneau (iD) http://orcid.org/0000-0002-8323-1511
Nicolas Buchon (iD) http://orcid.org/0000-0003-3636-8387

## Decision letter and Author response

Decision letter https://doi.org/10.7554/eLife.64125.sa1
Author response https://doi.org/10.7554/eLife.64125.sa2

# Additional files

## Supplementary files

• Supplementary file 1. Recipes of all diets utilized in this study.

- Supplementary file 2. R Markdown.
- Transparent reporting form

## Data availability

Data have been submitted with an ArrayExpress accession E-MTAB-10812.

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

## Appendix 1

### Appendix 1—key resources table

| Reagent type (species) or resource | Designation | Source or reference | Identifiers | Additional information |
|---|---|---|---|---|
| genetic reagent (*D. melanogaster*) | Canton-S | Bloomington *Drosophila* Stock Center | BDSC: 64349<br>FLYB: FBst0064349<br>RRID:BDSC_64349 | Flybase genotype: Canton-S |
| genetic reagent (*D. melanogaster*) | *Esg^TS* | **Micchelli and Perrimon, 2006** | N/A | w;Esg-Gal4; UAS-GFP, tub-Gal80$^{TS}$ |
| genetic reagent (*D. melanogaster*) | *Esg^TS* | **Nagy et al., 2018** | N/A | w;Esg-Gal4,tub-Gal80$^{TS}$,UAS-mcherry-CD8 |
| genetic reagent (*D. melanogaster*) | *Esg^F/O* | **Jiang et al., 2011** | N/A | w;*Esg-Gal4, UAS-GFP, tub-Gal80$^{TS}$; Act > STOP > Gal4,UAS-flp* |
| genetic reagent (*D. melanogaster*) | *Myo^TS* | **Buchon et al., 2010** | N/A | w$^-$; *Myo1A-Gal4, UAS-GFP, tub-Gal80$^{TS}$; upd3-lacZ* |
| genetic reagent (*D. melanogaster*) | *Act^TS* | This publication | N/A | *Actin5C-Gal4/Cyo; TubGal80ts,UasGFP* |
| genetic reagent (*D. melanogaster*) | 5,966GS | **Guo et al., 2013** | FLYB: FBti0150384 | *w-; 5,966GS/Cyo* |
| genetic reagent (*D. melanogaster*) | Single cell clonal system | **Ito et al., 1997** | N/A | *hsFlp; Act > STOP > Gal4,UAS-GFP* |
| genetic reagent (*D. melanogaster*) | ActGS | Bloomington *Drosophila* Stock Center | BDSC: 9431<br>FLYB: FBst0009431<br>RRID: BDSC_9431 | *P{ry[+ t7.2] = hsFLP}12,y[1] w[*]; P{w[+ mC] = UAS GFP.S65T}Myo31DF[T2]; P{w[+ mC] = Act5 C(-FRT)GAL4.Switch. PR}3/TM6B, Tb[1]* |
| genetic reagent (*D. melanogaster*) | *UAS-His-RFP* | **Mayer et al., 2005** | N/A | *UAS-Histone2B-RFP* |
| genetic reagent (*D. melanogaster*) | UAS-Gcn2-IR | Bloomington *Drosophila* Stock Center | BDSC: 67215FLYB: FBst0067215<br>RRID: BDSC_67215 | |
| genetic reagent (*D. melanogaster*) | UAS-LK6-IR | Bloomington *Drosophila* Stock Center | BDSC: 60003FLYB: FBst0060003<br>RRID: BDSC_60003 | |
| genetic reagent (*D. melanogaster*) | UAS-EGFR-IR | Bloomington *Drosophila* Stock Center | BDSC: 60012FLYB: FBst0060012<br>RRID: BDSC_60012 | |
| genetic reagent (*D. melanogaster*) | UAS-Tor-IR | Bloomington *Drosophila* Stock Center | BDSC: 34639FLYB: FBst0034639<br>RRID: BDSC_34639 | |
| genetic reagent (*D. melanogaster*) | UAS-Myc-IR | Bloomington *Drosophila* Stock Center | BDSC: 36123FLYB: FBst0036123<br>RRID: BDSC_36123 | |
| genetic reagent (*D. melanogaster*) | UAS-raptor-IR | Bloomington *Drosophila* Stock Center | BDSC: 34814FLYB: FBst0034814<br>RRID: BDSC_34814 | |
| genetic reagent (*D. melanogaster*) | UAS-S6k-IR | Bloomington *Drosophila* Stock Center | BDSC: 41702FLYB: FBst0041702<br>RRID: BDSC_41702 | |

*Appendix 1 Continued on next page*

*Appendix 1 Continued*

| Reagent type (species) or resource | Designation | Source or reference | Identifiers | Additional information |
|---|---|---|---|---|
| genetic reagent (*D. melanogaster*) | UAS-SREBP-IR | Bloomington *Drosophila* Stock Center | BDSC: 34073FLYB: FBst0034073 RRID: BDSC_34073 | |
| genetic reagent (*D. melanogaster*) | UAS-RagA-B-IR | Bloomington *Drosophila* Stock Center | BDSC: 34590FLYB: FBst0034590 RRID: BDSC_34590 | |
| genetic reagent (*D. melanogaster*) | UAS-RagC-D-IR | Bloomington *Drosophila* Stock Center | BDSC: 32342FLYB: FBst0032342 RRID: BDSC_32342 | |
| genetic reagent (*D. melanogaster*) | UAS-Atg8a-IR | Bloomington *Drosophila* Stock Center | BDSC: 34340FLYB: FBst0034340 RRID: BDSC_34340 | |
| genetic reagent (*D. melanogaster*) | UAS-Atg2-IR | Bloomington *Drosophila* Stock Center | BDSC: 34719FLYB: FBst0034719 RRID: BDSC_34719 | |
| genetic reagent (*D. melanogaster*) | UAS-Ras85D$^{V1}$ | Bloomington *Drosophila* Stock Center | BDSC: 64195FLYB: FBst0064195 RRID: BDSC_64195 | |
| genetic reagent (*D. melanogaster*) | UAS-Tor-DER | gift from M. Freeman | N/A | yw; Pw+; (UAS torD-DER) II RJH430, |
| genetic reagent (*D. melanogaster*) | UAS-upd3-OE | gift of M. Crozatier **Brown et al., 2001** | N/A | |
| genetic reagent (*D. melanogaster*) | UAS-spi-SEC | Bloomington *Drosophila* Stock Center | BDSC: 58436FLYB: FBst0058436 RRID: BDSC_58436 | |
| genetic reagent (*D. melanogaster*) | UAS-Rheb-OE | Bloomington *Drosophila* Stock Center | BDSC: 9690FLYB: FBst009690 RRID: BDSC_9690 | |
| genetic reagent (*D. melanogaster*) | UAS-rpr-OE | Bloomington *Drosophila* Stock Center | BDSC: 5823FLYB: FBst005823 RRID: BDSC_5823 | |
| genetic reagent (*D. melanogaster*) | UAS-InR-DN | Bloomington *Drosophila* Stock Center | BDSC: 8252FLYB: FBst008252 RRID: BDSC_8252 | The 8,252 line was backcrossed six times into an outbred population with a w$^{-Dah}$ background |
| genetic reagent (*D. melanogaster*) | GBE-Su(H)-lacZ | **Micchelli and Perrimon, 2006** | N/A | |
| genetic reagent (*D. melanogaster*) | thor-lacZ | Bloomington *Drosophila* Stock Center | BDSC: 9558FLYB: FBst009558 RRID: BDSC_9558 | |
| genetic reagent (*D. melanogaster*) | DGRP panel | **Mackay et al., 2012** | N/A | Lines used are reported in *Figure 1— source data 2* |
| chemical compound, drug | RU486 | Sigma-Aldrich | Cat# M8046;Puchem#: 24278572 | 100 µL of a 5 mg/mL solution of RU486 per vial |
| chemical compound, drug | FD&C1 blue dye | VWR | Cat# 700010–048; MDL# MFCD00012141 | |
| chemical compound, drug | Puromycin | Sigma Aldrich | Cat# P8833;MDL# MFCD00012691; PubChem# 24898984 | |
| chemical compound, drug | Trizol | Life Technologies | Cat#15596018 | |
| other | Yeast | MP biomedicals | Cat# 903,312 | |

*Appendix 1 Continued on next page*

*Appendix 1 Continued*

| Reagent type (species) or resource | Designation | Source or reference | Identifiers | Additional information |
|---|---|---|---|---|
| other | inulin | VWR | Cat# CAAAA18425-09 | |
| other | cellulose | VWR | Cat# IC19149980 | |
| other | pectin | Sigma Aldrich | Cat# P9135 | |
| other | lard | Walmart | Land o lakes | |
| other | casein | Sigma Aldrich | Cat# C5679 | |
| other | AA mix | Harlan Laboratories, *Lee and Micchelli, 2013* | Cat# TD.10473 & TD.110036 | |
| other | vitamin and mineral mix | Harlan Laboratories; *Lee and Micchelli, 2013* | Cat# TD.10475 | |
| other | cholesterol | Sigma Aldrich | Cat# C8667 | |
| other | moldex | Neta Scientific | Cat# QB-A611-0572-159 | |
| other | yellow cornmeal | Walmart | Aunt Jemima | |
| other | glucose | Alfa Aesar | Cat# A16828 | |
| other | maltose | Sigma Aldric | Cat# M5885 | |
| other | fructose | Sigma Aldrich | Cat# F0127 | |
| other | arabinose | VWR | Cat# 80502–266 | |
| other | sorbitol | VWR | Cat# 76177–308 | |
| other | nutri-fly BL cornmeal | Genesee scientific | Cat# 66–112 | |
| other | BL Molasses | Genesee scientific | Cat# 66–116 | |
| other | DAPI | Sigma-Aldrich | Cat# D9564 | 1:50,000 of a 10 mg/mL stock solution |
| other | Citifluor AF1 | Electron Microscopy Sciences | Cat #17970–100 | |
| antibody | Anti-pH3 (Rabbit polyclonal) | EMDMillipore - Sigma | Cat# 06-570 RRID:AB_310177 | IF 1:1,000 |
| antibody | Anti-pH3 (Mouse monoclonal) | EMDMillipore - Sigma | Cat# 05-806 RRID:AB_310016 | IF 1:1,000 |
| antibody | anti-β-Galactosidase (Rabbit polyclonal) | Invitrogen | Cat# A11132 RRID:AB_221539 | IF 1:1,000 |
| antibody | Anti-Prospero (Mouse monoclonal) | DSHB | Cat# MR1A RRID:AB_528440 | IF 1:100 |
| antibody | Anti-Mesh (Rabbit polyclonal) | Gift from Mikio Furuse, *Izumi et al., 2012* | 995-1 RRID:AB_2568117 | IF 1:2000 |
| antibody | Anti-peIF2α (Rabbit monoclonal) | Cell signaling technologies | Cat# 3398 RRID:AB_2096481 | IF 1:500 |
| antibody | Anti-puromycin (mouse monoclonal) | DHSB | Cat# PMY-2A4 RRID:AB_2619605 | IF 1:100 |
| antibody | Anti-mouse Alexa 555 (Donkey polyclonal) | Thermo Fisher | Cat# A31570 RRID:AB_2536180 | IF 1:2000 |

*Appendix 1 Continued on next page*

*Appendix 1 Continued*

| Reagent type (species) or resource | Designation | Source or reference | Identifiers | Additional information |
|---|---|---|---|---|
| antibody | Anti-rabbit Alexa 555 (Donkey polyclonal) | Thermo Fisher | Cat# A31572 RRID:AB_162543 | IF 1:2000 |
| antibody | Anti-mouse Alexa 647 (Donkey polyclonal) | Thermo Fisher | Cat# A31571 RRID:AB_162542 | IF 1:2000 |
| antibody | Anti-rabbit Alexa 647 (Donkey polyclonal) | Thermo Fisher | Cat# A31573 RRID:AB_2536183 | IF 1:2000 |
| commercial assay or kit | QuantSeq 30 mRNA-Seq Library Prep Kit FWD | Lexogen | Cat#015.2 × 96 | |
| software, algorithm | Fiji | *Schindelin et al., 2012* | RRID# SCR_002285 | |
| software, algorithm | OsiriX DICOM viewer | *Rosset et al., 2004* | RRID# SCR_013618 | |
| software, algorithm | Adobe Premiere pro | Adobe | RRID# SCR_021315 | |
| software, algorithm | Adobe Photoshop | Adobe | RRID# SCR_014199 | |
| software, algorithm | Adobe Illustrator | Adobe | RRID# SCR_010279 | |
| software, algorithm | Rstudio | RStudio Team (2020). RStudio: Integrated Development for R. RStudio, PBC, Boston, MA URL http://www.rstudio.com/. | RRID# SCR_000432 | |
| software, algorithm | FCS express 6 | De Novo software | RRID:SCR_016431 | https://denovosoftware.com/ |
| software, algorithm | FastQC | Babraham Bioinformatics | RRID:SCR_014583 | https://www.bioinformatics.babraham.ac.uk/index.html |
| software, algorithm | Cutadapt | *Martin, 2011* | RRID:SCR_011841 | |
| software, algorithm | TopHat | *Trapnell et al., 2009* | RRID:SCR_013035 | https://ccb.jhu.edu/software/tophat/index.shtml |
| software, algorithm | HTSeq | *Anders et al., 2015* | RRID:SCR_005514 | https://htseq.readthedocs.io/en/master/ |

