## [Decision Letter]

**Acceptance summary:**

This tour-de-force study provides a granular, quantitative, cell-to-organ scale analysis of dietary impact on tissue homeostasis and morphological plasticity in the *Drosophila* adult intestine. The authors systematically and dynamically vary dietary composition and assess the relative balance of stem cell proliferation and differentiated cell loss, cell and organ morphology, and response to niche signaling. The data presented give a highly detailed view of how the intestine dynamically responds to dietary change, leading the authors to a provocative interpretation that tissue homeostasis is a metastable state.

**Decision letter after peer review:**

Thank you for submitting your article "Diet composition resizes the fly midgut by affecting stem cell-niche coupling and enterocyte gain, loss, and size" for consideration by *eLife*. Your article has been reviewed by 3 peer reviewers, and the evaluation has been overseen by Lucy Erin O'Brien as Reviewing Editor and Reviewer #1, and Utpal Banerjee as the Senior Editor.

Summary:

The intestine's adaptive response to dietary change is conserved throughout the animal kingdom. This manuscript provides an in-depth examination of the nutritional inputs and cellular elements that mediate diet-driven adaptation of the adult *Drosophila* midgut; its holistic scope and rigorously quantitative approach make it one of the most comprehensive examinations to date and provide new insights that will be of wide interest.

All three reviewers recognized the overall substantial contributions of this work; at the same time, they had concerns about certain experimental conditions and particular aspects of data presentation and interpretation.

Essential revisions:

(1) Tor/autophagy experiments. Given that the myo1A driver expresses in both enterocytes and neurons, it is important to repeat at least one key Tor/autophagy experiment using a different enterocyte driver to confirm that the observed effects are due specifically to genetic manipulation of enterocytes. In addition, inclusion of Tor gain-of-function data or a readout of endogenous Tor activation in HY vs HS conditions would strengthen the authors' conclusions, if such data can be readily obtained; in light of *eLife*'s resubmission policies during COVID, however, these gain-of-function experiments are not essential.

(2) Homeostasis. Two Reviewers had concerns that the replacement ratio experiments, which are the basis of the authors' conclusions regarding homeostasis, were performed under physiological conditions that have been shown not to be homeostatic. In addition, one Reviewer had questions regarding the robustness of cell labeling and the formulas used to calculate new and old cells. Please address these comments in the manuscript text and/or response to reviews. Additional controls for the cell labelling, if they are available, would help to validate the assay (but are not essential).

(3) Diet manipulations. Please address the following reviewer comments, either in the text/response to reviews or by providing additional data. (a) Effect of gut capacity was not taken into account for excreta-based measurements of food ingestion. (b) Effects of diet on cell-type proportions in this study appear to differ compared to Obniski Dev Cell 2018. (c) Potential alternative interpretations of "antagonistic" effect of sugar on protein.

(4) Some improvements for clarity and context. I invite the authors to consider the Reviewers' comments with regard to (a) clarifying the key takeaways for a broad audience; (b) including crucial experimental details, such as animal age and mating status, in the Results (not just the Methods); (c) improving the supplemental tables that contain the diet recipes.

*Reviewer #1:*

In this tour-de-force study, the authors have used an impressively thorough, deconstructed, quantitative approach to examine the resizing plasticity of the adult *Drosophila* midgut in response to different dietary compositions and the underlying cellular behaviors that drive these physical changes. Specifically, when comparing flies raised on a high yeast (HY) or a high sucrose (HS) diet, they found that a HY diet increased midgut length, while a HS diet appeared to antagonize this effect. This diet-dependent organ resizing involves changes in cell number and cell size, and is both reversible and repeatable throughout the animal's adult life. HS promotes growth pathway gene expression, but without increased ISC proliferation. However, ISC proliferation alone cannot explain HY-induced midgut length increase. Rather, increased EC size, through TOR signaling, is required to increase midgut length, as blocking TOR signaling prevents midgut growth upon shifting from HS to HY diets.

This unprecedented work provides a highly granular, multi-dimensional understanding of diet-driven organ size plasticity. It examines multiple aspects that are more typically considered only in isolation (e.g. nutritional geometry, GWAS analyses, whole-organ morphometry/allometry, total counts of individual cell types, RNA seq analyses, detailed time courses) and synthesizes them into a comprehensive amalgamation of resizing phenomena. Along the way, the authors arrive at a number of important and surprising conclusions that will reverberate in the midgut field and in adult stem cell and tissue biology more generally. Further, the often complex data was superbly presented in creative and clear graphics, and the text was a true pleasure to read.

It should be noted that while the calculation of a "replacement ratio" is an illuminating and original approach to quantify cell turnover, there are three important elements that might deserve consideration. Specifically, the authors' calculations would benefit from further clarifications. Furthermore, there seems to be no evidence for highly efficient and uniform His::RFP retention, two essential preconditions for their replacement assay. Finally, the authors' interpretation that homeostasis may be merely "apparent" rests on an unaddressed, and potentially incorrect, assumption that replacement ratios are be constant throughout the animal's lifetime, and does not consider the possibility that guts are likely initiating age-associated dysplasia during the chase period.

Major comments:

1. The calculation of a "replacement ratio" is an illuminating and original approach to quantify cell turnover. The authors' findings that replacement ratios were significantly different from 1 for all conditions that they tested could be paradigm-challenging--if it is correct.

While I feel that this analysis was thoughtfully approached, there are three important elements that I either did not understand or else feel deserve consideration. Details are below. In brief: (a-d) I was confused by some parts of the authors' calculations. (e) I am concerned that there was no evidence for highly efficient and uniform His::RFP retention, two essential preconditions for their replacement assay. (f) Finally, I feel that the author's interpretation that homeostasis may be merely "apparent" rests on an unaddressed, and potentially incorrect, assumption that replacement ratios are be constant throughout the animal's lifetime, and does not consider the possibility that guts are likely initiating age-associated dysplasia during the chase period.

a. If I am understanding the analyses correctly, then it seems that gut size measurements do not support the calculated replacement ratios. For instance, the HY ratio from labeling days 0-14 (days 10-24 post-eclosion) is 1.64--this would imply that the gut should double in size every ~15 days. But the HY gut size, as measured by the authors, stays nearly the same over days 7-21 (Figure 4S1C) and only increases modestly by day 28. Can the authors explain these differences?

b. I was confused by the actual formula described in the Methods. For instance, the authors state, "The number of cells lost that were initially unmarked was calculated by first calculating the percentage of RFP+ cells present after 14 days (initial RFP+ number/ final RFP+ number). Shouldn't the % of RFP+ cells present after 14 days be the final/initial RFP+ number? As another example, "The number of cells lost that were initially unmarked (initial RFP- * (initial RFP+ number/ final RFP+ number)) was then calculated." I don't understand why the product of initial RFP- and the ratio of initial/final RFP+ cells yields the numbers of RFP- cells that were lost over time. Can the authors clarify their formula?

c. More generally--since all new cells are RFP-, I am having trouble grasping how one can count the numbers of initial RFP- cells that are lost without assuming that cell addition = cell loss (which is what the authors are refuting)?

d. The authors state that "not 100% of polyploid cells was marked at day 0". What % of polyploid cells were unmarked at day 0? If RU486 is administered for a longer period, will all polyploid cells acquire labeling (i.e. have they saturated the labeling potential)?

e. To use a label-retention protocol in this highly quantitative manner, two preconditions must be true: that (1) label retention is highly efficient virtually all cells labelled with HisRFP at day 0 retain the label 14 days later, and that (2) initial labelling and label retention are the same in HS and HY conditions. The images presented by the authors in 4G-J do not validate that these preconditions are met. They show that cells are capable of retaining the label, but they do not measure how many cells retain the label or whether labeling and retention are not altered by diet. Can the authors provide evidence for diet-insensitive and efficient label retention over the course of their experiments? If not, then this major caveat should be discussed in the text and the conclusions tempered accordingly.

f. Replacement ratio are likely not constant over the chase period that was used in these experiments (10-24 days post eclosion). One might argue that when the chase period begins (day 10 post eclosion), *Drosophila* adults are already "past their prime" (peak egg laying is ~4-7 days post eclosion), and that during the chase period the guts are undergoing age-associated dysplasia, which is morphologically apparent by 24 days (on a "standard" fly diet). Indeed, some prior evidence (Liang 2017 Figure 1) would support the notion that replacement ratio in early adults (4-8 days post eclosion) is indeed close to 1. Have the authors examined replacement ratios for chase periods that initiate earlier (e.g. 6-7 days post eclosion) and/or are shorter in duration (e.g. 2-4 days)? If not, then the possibility that replacement ratios are not constant, and that the ratios measured in Figure 4 may be affected by age-associated dysplasia, should be discussed.

2. The interpretation of the Figure 5, S51, S52-associated experiments strikes me as overstepping some limitations of the data to discount the role of progenitors (see a-c below). I do not feel that additional experiments are needed because the study's main contribution is the organ-scale and ECs analyses and not ISCs. However, the authors may wish to consider tempering the text of the Results to allow for alternative interpretations regarding the ISC responses, similar to the more equitable treatment that ISCs receive in the Discussion.

a. The conclusions about stem cell divisions rely on one sole readout of stem cell activity in the paper: PH3 staining. This measurement is a snapshot of cells in mitosis at a single point in time. whether those cells are cycling faster or slower is unknown. In addition, PH3 staining belies richer and potentially relevant information about stem cell division behaviors, such as progeny produced per stem cell and sibling fate outcomes, which can be revealed through genetic lineage tracing.

b. The conclusion that stem cells are "uncoupled" from growth signals by inhibition of translation in stem cells under HS conditions would require multiple additional experiments to be convincingly demonstrated. As one alternative explanation is that the growth signals themselves, despite being transcribed, are not translated, secreted, or otherwise available to signal to the stem cells.

c. The authors do not consider that enteroblasts represent an additional potential "node" for organ size control--they can differentiate at different rates, or even undergo apoptosis as reported recently (Reiff, 2019).

*Reviewer #2:*

This is an exciting manuscript that describes and characterises differential effects of high-yeast (HY) and high-sugar (HS) diets on migdut length in adult *Drosophila*. Through a series of carefully controlled experiments, the authors establish that a combination of various yeast-derived nutrients promotes midgut growth, whereas sugar antagonises yeast-induced growth independent of other confounds (e.g. effects on food intake). As well as confirming previous effects of diet on intestinal stem cell (ISC) proliferation, the authors uncover an additional layer of diet-dependent ISC regulation by identifying a role for HS-induced translational inhibition in restraining ISC proliferation. They also identify enterocyte (EC) size as a better predictor of the genetic/physiological variability in midgut size, and uncover a role for EC size in mediating midgut size plasticity. Mechanistically, they suggest that the diet-dependent regulation of EC size is mediated by the TOR/autophagy pathway.

Collectively, the experiments described in this manuscript advance our understanding of the nutritional plasticity of the adult intestine at the molecular, cellular and whole-organ levels. There are only a few conclusions that require additional clarifications and/or stronger data to support them. These concern previously reported effects of lipids, the physiological role of TOR signalling, possibly dietary effects on enterocyte ploidy and stem cell division mode, and the quantifications of food intake.

1. The HS and HY diets in this study differ in lipid composition. A recent study showed that, during the initial period of postnatal gut growth, dietary cholesterol modulates the number and fraction of enteroendocrine (EE) cells via its effects on Notch signalling (PMID: 30220569). In this manuscript, cell type ratios seem unchanged in the two diets (Figure 3S1A), which seems at odds with this study. Would the authors agree? I wondered whether the partial supplementations in Figure 2B did affect EE/EC ratios, perhaps in contrast to the full HY diets?

2. All of the experiments that address contributions of TOR signalling/autophagy used the MyoIA-Gal4 driver, which is expressed in ECs, but also in quite a few neurons including enteric neurons. At least one of these experiments should be repeated with mex-Gal4 or another EC-specific driver. Also, the loss-of-function data provided so far suggests that TOR signalling can mediate the diet-dependent difference in EC size, but not necessarily that it does so. A TOR signalling gain-of-function experiment in the context of gut size, and/or demonstration that the TOR signalling status of EC normally differs between HS-and HY-fed flies, would strengthen the case for this pathway as a physiological mediator of this difference. Related to this, I am not sure that the description of the "TOR/autophagy pathway as a master regulator" in line 545 is warranted either way. What do the authors mean by master regulator in this context?

3. Also related to the previous point, the authors do not seem to have considered possible differential effects of the diets on EC ploidy and/or ISC division mode (symmetric vs asymmetric). Do the larger enterocytes have increased ploidy? Do they become larger as a result of overall cell growth/ploidy or increased cell volume/dilution?

4. The authors carefully consider possible differences in caloric intake/content between the two diets: they use isocaloric diets, and attempt to rule out differences in food intake between the two diets. However, the use of excreta as a proxy for the latter is only justified when gut capacity remains unchanged (PMID 21195352), which is unlikely to be the case in their study where differences in gut lengths are apparent. A simple way around this is to quantify ingested dye colorimetrically following ingestion of a short pulse of dye-laced food (see, for example, PMID: 33116314).

Suggestions/specific points

1. I realise that elucidating how exactly sugar antagonises the effect of yeast is beyond the scope of this particular manuscript but, alongside the various sugars tested in Figure 2D, it would have been very informative to test the effects of a palatable, but not nutritious, sugar (e.g. arabinose) vs a nutritious, but not palatable sugar (e.g. sorbitol). This would help distinguish between sensory vs metabolic mechanisms.

2. The manuscript clearly shows that the two diets differentially impact ISC proliferation, EC loss and the gut's transcriptional profiles. How are these three features related? Its seems to me that one intriguing possibility is that young and old ECs differ in their transcriptional identity and/or ability to handle different diets, and different diets result in different fractions of young vs old EC. EC physiology/transcriptional profile may also be related to EC size/ploidy. If the authors have a cellular readout for one of the genes transcription of which is diet dependent, a simple co-staining of the HisRFP or esgFO experiments may help resolve this, but otherwise I suggest that the authors at least discuss this idea, possibly at the expense of the Discussion section "Is midgut homeostasis only an apparent property?" section. I thought this was a weak discussion point because (1) most experiments in the manuscript were conducted during the postnatal gut growth period (ie non-homeostatic) and (2) both the HS and HY diets are "imposed", not particularly ecological, diets. Hence, we do not really know whether the size of the gut would be stable in truly wild-type flies with food choices.

3. The fact that mated females were used for all experiments is only mentioned in the Methods. It would make more sense to state this in the manuscript, and justify the rationale for this choice; the dietary plasticity of the midgut may well differ between males, virgin and mated females in light of other data including a recent study by the authors (PMID: 26216039, 32641829, https://doi.org/10.1073/pnas.2018112118).

4. Both the main text and figures could do with stating more clearly whether the data was acquired during the gut growth period post-eclosion and/or the "homeostatic" period that follows. I realise that the information is there but it is sometimes hard to find.

5. Line 1336-7: what do the authors mean by "midgut re-sizing is allometric"? Allometric relative to what? The statement in the main text make sense to me (e.g. diet changes the allometry of different gut regions), but it is less clear to me what this sentence means.

6. What vitamins/minerals were provided in the Harlan mix? I was unable to find the composition online based on the catalogue number provided in the Methods.

*Reviewer #3:*

The authors use length of gut as measurement to asses growth in response to diet composition. They also measure a large panel of fly strains. They assess the ration of sugar to yeast, and conclude that sugar has a strong antagonistic effect on growth. They also test by examining genome wide association data, and transcriptome analysis, and examine translational control, TOR pathway components.

There are also some concerns about how to define low growth versus antagonistic effects. There is clearly different growth rate with the sugar vs yeast diet, but to suggest that high sugar has antagonistic effect is not sufficiently convincing, because it is equally good to describe it as low growth rate, with lower nutrient say by dilution as one would expect. Nonetheless, the idea that sugar (sucrose, glucose etc), not just carbohydrates or polysaccharides, actually has an antagonistic effect is an attractive direction. One possible improvement will be to investigate further into the "antagonistic" effect of sucrose, first to convincingly show that it is not just a low growth condition, then to investigate possible cellular or molecular targets that antagonize the growth. The results from transcriptome data, and the involvement of TOR and translational control in cell size is not unexpected as these pathways have been shown to be involved in cell growth in ISCs and EBs before, and their function in ECs is therefore a confirmation of previous work.

Large amount of data and analyses are provided, but everything seems to contribute to something, making the conclusions ambiguous and no less confusing than the beginning. Substantial reorganization and rewriting may simplify the presentation and focus on their main idea for more convincing arguments.

line 147 – The CS flies results first mentioned should be directly integrated into other flies, because the CS results (31% longer, 44% wider) seem to be to the extreme ends when compare to all 188 lines examined (median 1.18). Perhaps better to shorten this section with focus on the population study, which is much more valuable than individual strains.

Line 168, stated that 184/188 had larger midguts, but then followed by 126 lines statistically significant. They should reconcile these statements.

Concerning HY or HS diet, the authors should give a better description of these compositions at the beginning of result section, and the justification. The exact composition is quite important as these experiments occupy a large portion of the initial experiments. The detail confuses me: P. 85, supp Experimental conditions: Table 1, sucrose for preexperiment is 40 g/L, High sugar diet is 168.1, low sugar diet is 74.1. Why does this low sugar diet still contain so high of sucrose, what is the reasoning behind this choice? Also, these tables in the supp are poorly formatted and many small prints are missing/shifted in the pdf and cannot be followed easily.

Yeast should also contain plenty of sucrose and polysaccharides, or precursors of sucrose. Has this been accounted for in their experiments comparing the amount of sucrose/yeast ratio?

Line 425, the use of Rasv12 to stimulate ISC over sugar diet perhaps is not fair. While there is some increase of ligands in their RNAseq data, there may not be high level enough to produce proliferative effect. Have they shown that there is activation of these pathways/target genes in ISCs under high sugar? At the end, it is probably better to use overexpressed ligands rather than Rasv12 to do this experiment.

Line 450, the lack of increased gut length after knockdown of translational inhibitor in sugar diet requires further investigation. It again suggested a delicate balance can affect the outcome, because knockdown of these translational inhibitors affects other cell or tissue parameters.

The combination of sucrose and yeast is widely used in flies, but isn't trehalose is the main balance of source of energy in flies and synthesized from glucose, and how that is calculated into their "isocaloric" argument of the diet used for the comparison?

---

## [Author Response]

Essential revisions:(1) Tor/autophagy experiments. Given that the myo1A driver expresses in both enterocytes and neurons, it is important to repeat at least one key Tor/autophagy experiment using a different enterocyte driver to confirm that the observed effects are due specifically to genetic manipulation of enterocytes. In addition, inclusion of Tor gain-of-function data or a readout of endogenous Tor activation in HY vs HS conditions would strengthen the authors' conclusions, if such data can be readily obtained; in light of eLife's resubmission policies during COVID, however, these gain-of-function experiments are not essential.

To address the first part of this revision point, we have now knocked down *Tor* and manipulated autophagy using another enterocyte driver, the hormone-inducible driver *5966GS*, to express RNAi against *Tor* and *Atg8a*. Both experiments recapitulate our previous findings with the *Myo^TS^* driver, confirming enterocytes as the relevant cell population (Figure 7—figure supplement 2D, E and main text lines 582-583 and 605-608, figure legend lines 1960-1968, material and methods lines 859-867).

Additionally, we have added quantitative data of the clone-tracing system (*hsFlp; Act>STOP>Gal4,UAS-GFP*) experiment to manipulate the TOR pathway and autophagy. We compared GFP^+^ clones (expressing RNAi or overexpression constructs) to regular GFP^-^ cells in the same midgut. The results corroborate our initial images. We have added these quantifications to Figure 7D for *Tor-IR* and *Rheb-OE* and Figure 7J for *Atg2-IR*. We have also replaced the original images to better reflect the quantification (Figure 7B-C for *Tor-IR* and *Rheb-OE* and Figure 7I for *Atg2-IR)*. We also made changes in figure legend at lines 1671 and 1679 – 1680 and in material and methods at lines 1110 to 1113.

We have also conducted new experiments to test the physiological regulation of TOR pathway by our experimental diets, by quantifying in vivo levels of a *4ebp* reporter (Thor-lacZ transcription reports Foxo activity and is inversely correlated to TOR activity). These results align with the signal from our prior RNAseq experiment, corroborating the response of TOR to experimental diets. Figure 7—figure supplement 2A-C. Changes in text can be found at lines 565-567 (main text), and lines 1959 – 1960 (sup. figure legend). We also changed TOR/autophagy is a master regulator to “an important regulator” (line 607-608).

Overall, these results confirm our previous findings and strengthen the conclusion that TOR pathway activity in enterocytes partly mediates midgut resizing by diet.

(2) Homeostasis. Two Reviewers had concerns that the replacement ratio experiments, which are the basis of the authors' conclusions regarding homeostasis, were performed under physiological conditions that have been shown not to be homeostatic. In addition, one Reviewer had questions regarding the robustness of cell labeling and the formulas used to calculate new and old cells. Please address these comments in the manuscript text and/or response to reviews. Additional controls for the cell labelling, if they are available, would help to validate the assay (but are not essential).

We have tackled these concerns both through new experiments and changes in the text.

2a: concern regarding homeostatic conditions

To address the first concern that the experiment was conducted in non-homeostatic conditions, we have performed the cell loss assay in two new conditions:

– First, we have repeated the experiment shown in Figure 4G to L, but dissected flies at an earlier timepoint (5 days after shift, for a total age of 15 days post eclosion, instead of 24 days as the experiment presented initially in the first submission of manuscript). We have added this result in Figure 4—figure supplement 2C and in main text at lines 373-377. We have also included here (Author response image 1) these new data shown in the same format as Figure 4L (showing replacement ratios and rates of cell loss and gain). For this new set of data, we had to perform corrections to the chart, since there was no detectable cell loss, and in some cases we had slightly more Histone RFP^+^ cells counted at 5 days after induction, as expected from a noisy biological system. For panel 1 illustration, we corrected the data to show 0 cell loss. As shown in Author response image 1, even at 5 days post chase, the ratios are reminding of what we observe at the later timepoint for the various dietary conditions.

**Author response image 1. sa2fig1:** Data shown as rate relative to experiment start (cell /initial EC/ day) for Figure 4-figure supplement 2C (5 days post chase start, 15 days from eclosion)*.*</Author response image 1 title/legend>.

In order to strengthen our message, we have modified the main text at lines 406-409 and the figure legend at lines 1849-1852 to reflect the addition of this new panel in the manuscript.

Second, we have utilized the approach suggested by reviewer #1, to perform a cell loss assay in younger adults. In order to do so, we moved flies at eclosion for 1 day on either HS or HY diet. After this one day, we moved the flies on food with added RU486 for 3 days (until 4 days post eclosion). We then dissected flies at this first time point and removed flies from the hormone to start the chase. We then dissected flies 5 days post chase start, at 9 days post eclosion. Again, we find an increase in the size of the gut in this time frame for HY to HY, coinciding with an increase in total cell number, while HS to HS flies stayed at around the same size and experienced limited cell proliferation, as inferred by the number of blue cells. We have not included this data in the manuscript, as this dataset has a different experimental design, but please find the results here in Author response image 2.

**Author response image 2. sa2fig2:** Number of ECs in the posterior midgut, both marked (Red, old ECs) and unmarked (Blue, new ECs) by RFP, error bars are SE from 3 repeats, for the experiment described above.

Additionally, we have investigated whether, in our experimental conditions, the gut is in a state of dysplasia. To do so, we have surveyed the state of the midgut with a *Esg^TS^>UAS-RFP* at the timepoints presented in the original manuscript for beginning and end of chase. Flies were kept at room temperature (as the histone RFP experiments) and shifted at 29°C 3 days before dissection to activate the system. Overall, we do not see accumulation of progenitors, or progenitors with apparent abnormal shapes in any of these conditions (Author response image 3) . We therefore conclude that we could not find dysplasia apparent at these timepoints.

**Author response image 3. sa2fig3:** Esg^TS^>UAS-RFP at the start of the chase (Day0) vs at the end of it (Day14) for region 4 of the midgut. Scale bars are 50 µm.

Overall, while we agree with reviewer #1 that the replacement ratio is not likely to be constant throughout the lifespan of the flies, as also indicated by our proliferative data in Figure 4B and previous publications, and that it is likely than on certain diets and at certain age there may be a 1:1 replacement ratio leading to homeostasis, as shown for example in Liang 2017 (10.1038/nature23678), on the diets in our manuscript we observe a non 1:1 replacement ratio also at earlier time points, suggesting that our results are not a consequence of accelerated renewal due to dysplasia.

We would also like to answer here a comment from reviewer #2, that the “Is midgut homeostasis only an apparent property?” section may be weak because most experiments were conducted during the post-natal growth period. While many experiments in the manuscript are indeed performed in the initial post-natal growth period, most of the experiments to survey cell gain and cell loss are made at later time points (and now also at earlier timepoints). In addition, we show that the increase in size or shrinking occur plastically, both early on and later in life. We therefore believe that we capture cell dynamics over a broad age range and cover the range usually considered as perfectly homeostatic. We agree that it may confusing to keep track of the time at which each experiment was performed only using the information in material and methods, as remarked also by reviewer #1, so we have added in the figure legends the age of the flies for each experiment.

Additionally, while the focus of the comments has been on the HY diet, since it is the one more resembling previously used “standard” conditions, the HS diet also does not show a 1:1 replacement condition, suggesting that indeed there is not a constant coupling between cell loss and gain, but instead a flexible connection between ISC proliferation and cell loss as a function of diet, contributing to midgut size. Regarding this finding, reviewer #2 pointed out that both HS and HY diets are “imposed” and not particularly ecological diets, and hence we do not know whether the size of the gut would be stable in wild-type flies with food choices. We agree with the reviewer that the experiments in this manuscript are not relevant to an ecological context, but they are designed to study the effect of specific diets and ratio of nutrients. We have added in the discussion a sentence to highlight this point at lines 731-736 of the discussion.

2b: concern regarding the robustness of our cell loss assay

As our cell chasing system is indeed an important component of the manuscript, we have also added details to demonstrate the robustness of the *5966GS* mediated cell labelling system. First, we would like to indicate that the same *His-2BRFP* that we used in our manuscript, has been previously used in at least two publications in the *Drosophila* midgut: PMID 26077448, where it is indicated that it persists for at least 28 days in the intestine (albeit under a “data not shown” label) and PMID 33135280. Additionally, we decided to also test the robustness of the system directly. To this purpose, we utilized *ActGS>UAS- His-2BRFP* to drive expression in all tissues in *Drosophila*, so that we could assay the dye stability in tissues that do not undergo turnover like the midgut. In such tissues, the lack of His-2BRFP signal after a chase (the cell would be DAPI+ RFP-) would not be due to the fact that new cells appear, but because the RFP is lost from these cells. We selected the crop and the hindgut for their proximity to the midgut and their lack of turnover in basal conditions. We utilized the same experimental settings as in Figure 4K. We observed high stability of the *His-2BRFP* in both these tissues, and in a similar manner on both diets. We have included this piece of data in Figure 4—figure supplement 2B. We have added a related description in the main text at lines 363-371, in the figure legend at lines 1840-1846 and in mat and met at lines 820-823.

Related to cell labelling, we agree with reviewers that not all cells are initially marked by His-RFP, in Figure 4K. This is in our hands the case of all enterocyte drivers: they drive in most but not all enterocytes. Importantly, in Figure 4K it is possible to see under the label HS and HY the number of cells that were initially marked (in red) or not marked (in blue). For guts on HS diet we counted an avg of 152 unmarked cells over a total of 1242 cells (12.2% of cells, so 87.8% of marked cells), meaning that the labelling coverage was high. For guts on HY diet we counted an avg of 258 cells unmarked over 1855 cells (13.9% of cells, so 86.1% of marked cells). As a corollary experiment to the new cell loss assay included for this revision, we also kept some flies for 5 days instead of 3 on RU486. We have obtained very similar percentage of cell marking, as it is shown in Author response image 4, suggesting that our experimental conditions already maximize the number of ECs labelled.

**Author response image 4. sa2fig4:** Percentages of RFP+ cells (RFP) over total number of cells (Dapi) at the start of a pulse chase experiment.

2c: concern about the calculation of replacement ratios

Regarding the formula to calculate cell loss, we would like to thank reviewer 1 for their comment, as we noticed that we had reported the formula incorrectly in the mat and met. We corrected the material and method section to reflect our calculations at lines 1069-1080. Importantly, we calculate the number of cells lost from the epithelium using solely RFP^+^ ECs and assume that this ratio of cell loss is similar also for the few unmarked cells.

Regarding the concern that a gut with a replacement ratio of 1.46 would double in size in 15 days, we think that while the gut changes how much it is turning over (as shown also in Figure 4B for proliferation), this does not (always) translate in a change in gut size (as visible in Figure 4B [pH3] and related 4S1C [length]). This result is in agreement with our finding in Figure 6 and 7 that the proliferative state of the gut is not the only, or even the central determinant of the final size of the organ. Overall, the total amount of cells for the posterior region changes of around 1000 cells (Figure 4K), but the size of these cells is likely to be slightly smaller (or with a different organization), leading to a comparable final size, which is nevertheless still bigger than the starting midguts. We have added a related comment at lines 406-409 to clarify.

Overall, we believe that the results from these experiments provide enough information to convince of the functionality of our EC loss assay, and therefore of our conclusions.

(3) Diet manipulations. Please address the following reviewer comments, either in the text/response to reviews or by providing additional data. (a) Effect of gut capacity was not taken into account for excreta-based measurements of food ingestion. (b) Effects of diet on cell-type proportions in this study appear to differ compared to Obniski Dev Cell 2018. (c) Potential alternative interpretations of "antagonistic" effect of sugar on protein.3a: concern about excreta based measurements

Regarding the concern about the excreta-based measurements, reviewer #2 is worried that the different sizes of guts on different diets would influence the amount of excreta due to different capacities. We believe that the design of our experiment (that lasts over multiple days) is such that we measure transit at a scale where the volume of one gut is neglectable. Precisely, we have taken our measurements over a 24-hour period, and it was previously published that the transit for the food is shorter, with most food transiting the gut in the first 2-3 hours from ingestion (https://doi.org/10.1128/mBio.01453-17). Additionally, we have taken our measurements by using the same flies consecutively for 5 days, taking a measurement every 24 hours (for a total of 3 repeats, and 5 timepoint for each repeat). This means that at each day but the first, the gut was already filled with blue food from the previous day. Since the measurements at each timepoint were extremely close to each other, we consolidated them in the manuscript. In order to convince the reviewer of this robustness, we present, in Author response image 5, an example of the time course for one diet belonging to the geometrical framework. We do see for some diets a slightly lower input in the initial day, in agreement with a possible role of gut content size. However, it becomes neglectable at day 2 and is not a concern as our assay lasts for 5 days total. We made changes in the material and methods to better describe the method used as lines 941-944.

**Author response image 5. sa2fig5:** Feeding assay showing data per each day instead of consolidated for one of the diets in the nutritional geometry experiment.

3b: concern about lipids and cell composition

Regarding the comment on differences in cell type proportion (reviewer #2), we do not think that our data is in contrast with the Obniski Dev Cell. While it is true that the amount of lipids is different between the HS and HY diets, there is a critical difference. While in the Obniski Dev Cell the amount of lipids is the only difference between the 2 diets used (addition or not of lipids), this is not the only change found between the HS and HY diets: we do not know about the effects of changes in lipid amount, on top of also varying amount of sugar and yeast. Additionally, in our own experiment, the composition of lipids between diets was the same, as lipids were provided by either high or low amount of yeast, while in the Obniski Dev Cell a cocktail of different lipids was added. Overall, we think we cannot compare these diets, and thus their effect on the ratio between EEs and ECs.

3c: concern about the interaction between sugar and yeast

We agree with reviewer# 3 that the interpretation of the interactions between sugar and yeast are difficult to untangle, but we believe our manuscript makes the point for an actual antagonism. We have several arguments that we believe convergently suggest an antagonistic role of sugar on yeast triggered growth, and we believe they demonstrate that the role of sugar goes beyond simply affecting the amount of yeast indirectly:

– First, our nutritional geometry approach in Figure 2 allows to map the contribution of sugar quantity, yeast quantity, and their relative amounts. Using this approach, we found that the more sugar was ingested, the smaller the gut would be, across different yeast to sugar ratios, and across different caloric contents. This means that for a given quantity of yeast ingested, the more sugar is ingested, the shorter the gut will be. This is evidence that the amount of yeast is not the only driver of gut size.

– Additionally, the experiment performed in Figure 2F shows that a low amount of yeast in the diet is enough to provide for gut growth if the rest of the calories is not provided by sugar, but by lipids. This demonstrates that sugar opposes the growth of the gut directly. Considering that yeast is indeed the main driver of gut growth, we believe this clearly shows the antagonistic role of sugar toward yeast induced midgut growth.

– Additionally, our transcriptomic study shows that the high sugar diet elicits a stress response in the gut. Regardless of the specific consequences of this stress, we think this “toxic effect of sugar” is clearly different from the effect of yeast on host transcriptome.

– Our mechanistic insights demonstrate that sugar directly uncouples ISC from their niche, thus decreasing cell proliferation, while yeast promotes proliferation. We have now expanded this conclusion to demonstrate that sugar probably affects the translation of proliferative signals in the niche, thus altering ISC-niche coupling. Specifically, to further investigate mechanisms through which HS diet blocks ISC proliferation via translation, we investigated in which cell type GCN2 mediated translational stress is important. We performed knock-down of *Gcn2* via RNAi in progenitor cells and in ECs respectively. We found that it was possible to increase proliferation only when knocking *Gcn2* in ECs, but not in progenitors (Figure 5—figure supplement 2G), suggesting that sugar induces stress in ECs, thus decreasing their ability to send pro-proliferative signals. This result also reinforces the idea that sugar does directly uncouple pro-proliferative signals and proliferation is in the niche. Changes in main text can be found at lines 498-502 and at lines 1889-1892 in the figure legend.

(4) Some improvements for clarity and context. I invite the authors to consider the Reviewers' comments with regard to (a) clarifying the key takeaways for a broad audience; (b) including crucial experimental details, such as animal age and mating status, in the Results (not just the Methods); (c) improving the supplemental tables that contain the diet recipes.

We have implemented suggested changes from reviewers, both by adding new results and through changes in text/organization. In this section of the response, we would also like to tackle comments that were not directly related to any of the other 3 main points but implemented in our revision as we believe they improve the manuscript.

4a Comments on proliferation and pH3

– Regarding the comments from reviewer #1 about the limitations of our study of proliferation and uncoupling. We agree with the reviewer that pH3 only reveals one facet of proliferation. In the descriptive part of our work, we demonstrate in detail the differences in proliferation between the HS and HY diet, using pH3, flip out tracing, and now our new cell gain and loss assay (5966GS His-RFP). So, we believe we make a strong case that all these indicators reflect a lack of proliferation on the HS diet. Nevertheless, we agree that when focusing on uncoupling between ISCs and their niche, we work with pH3 counts as a main phenotype, as we have established that this varies with diet, and as this is independent of Gal4 manipulation. Sadly, additional ways to monitor proliferation would be very complicated to combine with Gal4 manipulation of the translation machinery. Nevertheless, we have modified the text at lines 784-788 to highlight the possibilities suggested by the reviewer. In addition, we now have additional experiments that help clarify the role of progenitors in the response to diet:

– First, we demonstrate that transgene-based expression of pro-mitotic signals in ECs is sensitive to diet. In HS, overexpression of either *upd3* or spitz leads to less proliferation on HS than on HY (Figure 5E). In addition, when translation blockage is alleviated from ECs, but not progenitors, proliferation is rescued as measured by pH3 (Figure 5—figure supplement 5G). Altogether, these data suggest that the “uncoupling” we propose occurs really at the level of the ECs. We now clarify this in the text and agree with the interpretation of the reviewer about the uncoupling. We do think that what the reviewer described is the uncoupling, which acts through a problem in the ability of ECs to perform translation. We indeed think that these are the mechanisms through which uncoupling may happen between the niche and stem cells, hence the focus on translation in Figure 5. We have clarified this concept at lines 455-457.

– We have also reinforced our conclusion that over the period of time we focus on, progenitors are dispensable for gut growth. Ablation of progenitors using the pro apoptotic gene reaper (Figure 6—figure supplement B-F) shows that the gut is still able to grow in absence of progenitors (see later for more detailed comment on this experiment). However, we agree with the reviewer that enteroblasts represent a potential “node” for organ size control. Possibly, the ability for EC size to compensate for the lack of proliferation is not infinite and progenitors would reveal a more central role in longer studies. Since we did not include any experimental result about this point, we had left it out. We have now included it in the discussion at lines 766-769.

4b Comments on our diets in comparison to other diets

Reviewer #1 asked how our diets compared to standard diets used in the field of *Drosophila*. We have surveyed the effect of “standard” fly food diets on the length of the midgut. We have not included these diets directly in the nutritional geometry, as they also contain cornmeal and therefore would add additional complexity, but we have included this result in Figure1—figure supplement 1B and lines 152-154 in main text, and lines 1698-1699 in figure legend. This experiment showed that our HY diet has similar size to both Bloomington diets.

4c Diverse comments on minor points and confusing language and descriptions

– Regarding the comment on antagonism not being the best descriptor of the relationship between yeast and sugar: we changed antagonism to opposite throughout the manuscript.

– Regarding the comment that Lipid HS is a confusing name, we agree with the reviewer. Accordingly, to precise the diet composition and clarify our message, we changed Lipid HS to Yeast:Lipid 1:14 and Lipid HY to Yeast: Lipids 1:0.7. We did also change HS to Yeast: Sugar 1:14 (HS) and HY to Yeast:Sugar 1:0.7 (HY), and Lipids only to Yeast:Lipid 0:1. We hope this will make it more clear to understand which diet is what.

– As requested, we moved Figure2—figure supplement 1 D-F to the main figure 2 (now B, C, D) and re-arranged the figure and figure legend accordingly.

– As requested, we moved Figure3—figure supplement 1 A to the Figure 3 (now D) and re-arranged the figure and figure legend accordingly.

– As requested, we removed “continuously” from line 317. Guts do become significantly bigger at day 28 on HY and smaller on HS compared to the start of the experiment, so we kept the rest of the description.

– Regarding the comment asking about the practice (and rationale) of drawing lines on a PCA plot (Figure 5A), the lines describe the trajectory from eclosion of each diet. For each diet, they connect Day 0 to 1, then 1 to 2, 3 to 5. We briefly updated the figure legend at lines 1591 – 1593 to make this clearer. We also added the number of replicates (3) for the RNA-seq in the figure legend, as requested. For the part of the comment asking for single datapoints: for this visualization we prefer a single point for each sample, as we are not trying to show how much the samples from different repeats are close to each other as a mean of “quality control”, but to give a biological message on the changes and differences between eclosion and the 2 different diets. Additionally, a crossbar is present on the plot representing the standard error estimated from the three replicates.

– Regarding the comment on *Ras^V12^* overexpression (5D) having similar levels of proliferation as Canton-S in Figure 4B, we have included in the panel (5D) the control and an additional mean of eliciting proliferation, a *UAS-Tor-DER* construct. It is possible to see that the base level of pH3 stain in these flies is lower than in Canton-S, and that over-expression of both constructs with *Esg^TS^* results in a strong increase in pH3 cells. We made few changes in the main text at lines 460-463 to implement this change. We also precised the protocol used for the induction of pH3 with this method at line 853-857.

– Regarding the comment asking how midgut width was measured: we did measure width in the wider part of the anterior, middle and posterior midgut, as the reviewer mentioned similar to the 3 yellow lines perpendicular to the gut in Figure 1 C, D. We used the sum of these 3 widths for the plot. We updated description of midgut width calculation in Figure 1—figure supplement 1A at lines 1695-1698.

– We added a note about the plasticity experiment in O’Brien 2011 at lines 330-331 as requested.

– Regarding the comment asking for the differences between 5L, M vs figure supplement 5E, F. The experiment in 5L, M shows the data for guts kept on HS diet for 1 week after Gal4 activation. The data in Figure5—figure supplement 2E shows that data for guts that were first kept for 1 week on HY diet and the shifted-on HS diet. We have corrected an error in the palette in Figure5—figure supplement 2E, which was the one used for HS instead of HY to HS.

– Regarding the comment for the figure legend of Figure 6, that in the experiment where we drive EGFR-IR, depleted was too strong of a word, we changed description of effect of *Egfr* on ISCs in the figure legend at lines 1630-1632 to be more in phase with what was written in the results. We also performed an additional experiment, found in figure 6 —figure supplement 1 B-F, where we over-express *reaper* in progenitor cells with *Esg^TS^*. Over-expression of *reaper* result in loss of marked progenitor cells after 7 days at 29C (12 days from eclosion) in region 4 of the midgut (flies were on HS diet). Upon shifting these flies on HY, we observe a change in the morphology of the midgut similar to *Esg^TS^>Egfr-IR* (increased space between nuclei and lack of small cells). We also observe a change in the size of the midgut similar to the control, as observed with *Esg^TS^>Egfr-IR*, reinforcing our conclusion on the ability of the gut to change its size over a 7 day time period without the need for stem cells. Changes in main text can be found at lines 529-535 and 545-547, and at lines 1920-1931 in the figure legend.

– We added n numbers in Figure 6A and 7A as requested. Additionally, regarding the comment on why 7/14 sample being statistically correlated is a “generally positive correlation”, the generally positive correlation is more visible for HY (added a note for this in the figure legend), since HS diet do not vary in size as much and it is harder to observe any phenotypes there. This effect is also enhanced when comparing this result with the results in Figure 6A, which show a marked lack of correlation.

– We double checked all the references in the main text to figures to make sure they properly coincide, thanks for noticing. We also double checked the bibliography for mistakes.

– Regarding the comment on the title not doing justice to the manuscript, we renamed it to: Multiscale analysis reveals that diet-dependent gut plasticity emerges from alterations in both stem cell niche coupling and enterocyte size.

– Regarding the comment from reviewer#2, that it would be interesting to elucidate the role of sugar in antagonizing the effect of yeast, in particular to distinguish sensory vs metabolic mechanisms, we do agree and thank the reviewer for their comment. We followed the reviewer recommendations, and we have surveyed the effects of a palatable, but not nutritional sugar (arabinose) and of a non-palatable, but nutritious sugar (sorbitol). We found that using sorbitol, we have a similar response as sucrose, indicating that taste per se it is not required for the effect of sugar on the midgut. On the other hand, arabinose was lethal to the flies in just 3 days on the HS diet and reduced the overall size of the midgut on HY after 5 days. Considering the effect on HY diet, it is possible that arabinose is having some sort of stressful effects on the fly midgut or on organismal metabolism, thus leading to a shortened gut and a quick demise of the flies, rather than just being the lack of nutrition. We have added these data to Figure 2—figure supplement 2C, in the main text at lines 267-275 and in the figure legend at lines 1770-1774.

– Regarding the comment from reviewer#2 enquiring about possible changes in ploidy, we have now surveyed the impact of our HS and HY diet on cellular ploidy via FACS. Overall, we do not see an effect on ploidy due to HS and HY diet. We have added this result in Figure 3—figure supplement 1D-H, in the main text at lines 302-307, in material and method at lines 1028-1052, and in figure legend at lines 1803-1807.

– Regarding the comment of reviewer #2 on the possibility that young and old ECs differ in their transcriptional identity and capacity to handle diets/ challenges, we agree that it is indeed intriguing. Indeed, we have already two members of our lab working on this question! However, we believe this is beyond the scope of this publication.

– Regarding the comments on the importance of the sex/mating status of the flies on midgut growth, as also recently shown by us in a recent PNAS study, we agree with the reviewer that this is an important point to check. We have now added a panel in Figure 1—figure supplement 1 (new C) where we investigate the impact of HS and HY diets on also on un-mated females and males. Indeed, we found that mated females are the most responsive to diet compared to the other two conditions. We have made changes to the main text at lines 154-159, and in the sup Figure legend at lines 1699-1702.

– Regarding the question from reviewer#2 on “what do the authors mean by “midgut re-sizing is allometric?”, we changed the “midgut re-sizing is allometric” to midgut re-sizing is allometric between regions to clarify this point at line 1501.

– About the request for the composition of the vitamins/mineral mix, it is possible to find the composition in PMID: 23844001, File S1: Summary of food recipes used in this study (https://www.ncbi.nlm.nih.gov/pmc/articles/PMC3699577/bin/pone.0067308.s008.xlsx), CDF 100-500K Recipe tab.

– Regarding the comment from reviewer#3, that the part on Canton-S flies in Figure 1 should be shortened to focus on the DGRP, we would prefer to keep as it is, since we use Canton-S for many experiments in the manuscript, and we would prefer to have a clear representation of the phenotype in this fly line.

– Regarding the comment from reviewer#3 that “Line 168, stated that 184/188 had larger midguts, but then followed by 126 lines statistically significant. They should reconcile these statements”, we made changes when describing the DGRP to better convey our message. The first statement at lines 177-178 reflects that midguts on HY were quantitatively larger than midguts of flies on HS (ratio HY/HS was higher than 1 in 184/188 cases). The second statement at lines 179-181 is instead checking if the midguts were statistically different between HS and HY. We have also changed our statistical analysis to and because we noted an error in our tabulation of lines showing a statistically significant response to diet, which was previously read from a model summary (in error). We have remedied this error using post-hoc tests, see also changes at lines 958-961.

– Regarding the comment from reviewer#3 on “the rationale of the choice of HS and HY”, we used these diets based on a previous study (PMID: 25520356, now added to the manuscript ref.) for their diverging sizes despite being isocaloric.

– Regarding the comment from reviewer#3 on why the HY diet contains so much sugar, the idea was to study the effects of varying ratios of nutrients in an isocaloric context, not to have a sugar free or sugar rich comparison. In this context, the HY diet is not supposed to just be a low sugar diet, it is an isocaloric diet with a different ratio of Yeast:Sugar. We have also updated the results with this information at lines 144-146.

– Regarding the comment from reviewer#3 on the supplemental table being poorly formatted, we thank the reviewer. Formatting was lost when uploading the table as a csv file. We have now moved the tables from the uploaded csv file to a word document, for better formatting (now Supplementary File 1).

– Regarding the comment from reviewer#3 “if the amount of nutrients has been accounted when comparing”, this has been taken in account when calculating the amount of nutrients (see Figure 1B), but for nutritional geometry the ratios are based on grams of yeast and sugar used to cook the diets.

– Regarding the comment from reviewer#3 on *Ras^V12^* to stimulate ISC being “not fair”, we want to clarify that we used this construct to assay the proliferation capabilities of ISCs on HS condition, we were not trying to imply that *RAS^v12^* was the factor responsible for proliferation. Additionally, we checked the ability of stem cells to proliferate when expressing *UAS-upd3-OE* and *UAS-spi-SEC* from enterocytes via *Myo^TS^,* as suggested by the reviewer. We find that while both constructs can induce proliferation on HY, they are not able to do so on HS, suggesting again a role for the niche in the uncoupling between pro-growth signals and stem cells (Figure 5E). We update description in main text at lines 465-474, in figure legend at lines 1605-1607 and in Materials and methods at lines 857-859.

– Regarding the comment from reviewer#3 that trehalose is the main circulating sugar, and if this has been taken in account when calculating the isocaloric status of the diets: trehalose is indeed one of 2 circulating sugars in *Drosophila*, but the type of circulating sugar does not change the amount of calories ingested by the flies by eating sucrose in the HS and HY diets, even if nutrients are then converted to different components during digestion and metabolic processes, so we do not think this is a problem for the calculation of the isocaloric ratio of the diets.

– We have also added statistical brackets where applicable to figures, to ease recognition of what samples are being compared.